# Leveraging correlations between variants in polygenic risk scores to detect heterogeneity in GWAS cohorts

Jie Yuan[1]*, Henry Xing[1], Alexandre Louis Lamy[1], The Schizophrenia Working Group of the Psychiatric Genomics Consortium, Todd Lencz[2], Itsik Pe'er[1]

**1** Department of Computer Science, Columbia University, New York, United States of America, **2** The Center for Psychiatric Neuroscience, Feinstein Institutes for Medical Research, New York, United States of America

\* jyuan@cs.columbia.edu

**Data Availability Statement:** Data from all simulations and code to reproduce all figures in the manuscript and supplement are available on Github

## Abstract

Evidence from both GWAS and clinical observation has suggested that certain psychiatric, metabolic, and autoimmune diseases are heterogeneous, comprising multiple subtypes with distinct genomic etiologies and Polygenic Risk Scores (PRS). However, the presence of subtypes within many phenotypes is frequently unknown. We present CLiP (Correlated Liability Predictors), a method to detect heterogeneity in single GWAS cohorts. CLiP calculates a weighted sum of correlations between SNPs contributing to a PRS on the case/control liability scale. We demonstrate mathematically and through simulation that among i.i.d. homogeneous cases generated by a liability threshold model, significant anti-correlations are expected between otherwise independent predictors due to ascertainment on the hidden liability score. In the presence of heterogeneity from distinct etiologies, confounding by covariates, or mislabeling, these correlation patterns are altered predictably. We further extend our method to two additional association study designs: CLiP-X for quantitative predictors in applications such as transcriptome-wide association, and CLiP-Y for quantitative phenotypes, where there is no clear distinction between cases and controls. Through simulations, we demonstrate that CLiP and its extensions reliably distinguish between homogeneous and heterogeneous cohorts when the PRS explains as low as 3% of variance on the liability scale and cohorts comprise $50,000 - 100,000$ samples, an increasingly practical size for modern GWAS. We apply CLiP to heterogeneity detection in schizophrenia cohorts totaling > 50,000 cases and controls collected by the Psychiatric Genomics Consortium. We observe significant heterogeneity in mega-analysis of the combined PGC data (p-value $8.54 \times 0^{-4}$), as well as in individual cohorts meta-analyzed using Fisher's method (p-value 0.03), based on significantly associated variants. We also apply CLiP-Y to detect heterogeneity in neuroticism in over $10,000$ individuals from the UK Biobank and detect heterogeneity with a p-value of $1.68 \times 10^{-9}$. Scores were not significantly reduced when partitioning by known subclusters ("Depression" and "Worry"), suggesting that these factors are not the primary source of observed heterogeneity.

at the following link: https://github.com/jyuan1322/CLiP Due to patient confidentiality protections, genotype data of schizophrenia patients are available on request from The Schizophrenia Working Group of the Psychiatric Genomics Consortium (PGC) for researchers who meet the criteria for access to confidential data. For general instructions on data access, researchers may visit the PGC Data Access Portal at https://www.med.unc.edu/pgc/shared-methods/data-access-portal/. The Schizophrenia Working Group representative can be reached at pgc.dac.scz@gmail.com, and contact information for all members of the PGC Data Access Committee are available at https://www.med.unc.edu/pgc/about-us/people/data-access-committee/. Due to patient confidentiality protections, genotype data of neuroticism patients are available on request from the UK Biobank for researchers who meet the criteria for access to confidential data. Researchers may access data by registering in the UK Biobank Access Management System and submitting a proposal as outlined at https://www.ukbiobank.ac.uk/uk-biobank-access-management-system-ams-user-guide-getting-started/. The UK Biobank Access Management Team may be contacted at access@ukbiobank.ac.uk, with additional contact information available at https://www.ukbiobank.ac.uk/contact-us/.

**Funding:** T.L. is supported by grant R01 MH117646-02S1 from the National Institutes of Health. I.P. is supported by grants CCF-1547120 and DGE-1144854 from the National Science Foundation as well as grant U54CA209997 from the National Institutes of Health. The funders had no role in study design, data collection and analysis, decision to publish, or preparation of the manuscript.

**Competing interests:** The authors have declared that no competing interests exist.

## Author summary

Several traits, such as bipolar disease, are known to be heterogeneous and comprise distinct subtypes with unique genomic associations. For other traits such as schizophrenia, heterogeneity may be suspected, but specific subtypes are less well characterized. Furthermore, conventional mixture model-based methods to detect subtypes in genome-wide association data struggle with the high polygenicity of complex traits. We propose CLiP (Correlated Liability Predictors), a method that does not identify subtype-specific effects, but is very well-powered to detect heterogeneity of any kind within the very weak signals of GWAS. CLiP serves as a method to flag particular phenotypes for potential further study into the genomic factors driving heterogeneity, as well as a means to evaluate the transferability of polygenic risk scores. We also develop extensions of CLiP applicable to scoring heterogeneity in quantitative phenotypes and quantitative predictors such as gene expression. We apply CLiP to scoring heterogeneity in schizophrenia cohorts from the Psychiatric Genomics Consortium and neuroticism in individuals in the UK Biobank and find significant heterogeneity in both phenotypes, warranting further study.

## Introduction

In recent years Genome-Wide Association Studies (GWAS) have identified thousands of genomic risk factors and generated insights into disease etiologies and potential treatments [1, 2, 3]. Many GWAS apply logistic regression to case/control data to report SNPs with odds ratios of significant magnitude. An alternate formulation is the liability threshold model, which assumes case/control labels are sampled by thresholding a hidden quantitative polygenic risk score (PRS) with linear effects over SNPs [4]. This view of GWAS underlies a large body of work in predicting risks of disease using PRSs [3, 5] as well as quantifying the variance explained of the PRS [6, 7]. The logistic and liability models have been reported to be largely interchangeable by transforming log odds ratios to effects on the liability scale [8] and produce similar estimates of disease risk [9, 10]. Increasingly, there has been interest in advancing beyond these associations towards obtaining a deeper understanding the mechanisms by which genomic factors influence disease [1, 11]. These require models beyond simply combining linear effects of variants, as they often modulate phenotypes indirectly, through the expression of other genes [12, 13].

One such avenue has concerned the apparent heterogeneity of diseases which has not been sufficiently recognized by GWAS: while individuals in cohorts for these studies are frequently classified simply as cases or controls, clinical evidence for several GWAS traits have suggested that there are multiple different subtypes consisting of distinct sets of symptoms and association with distinct rare risk alleles [14, 15]. For example, polygenic risk scores for major depressive disorder explain more of the phenotypic variance when cases are partitioned into two known subtypes (typical and atypical), and the two subtypes exhibit polygenicity with distinct traits [16]. Similarly, by separating bipolar disorder into its two known subtypes corresponding to manic and hypomanic episodes, distinct polygenic risk scores comprising different associated SNPs are discovered, with genetic correlation being significantly lower than when individuals are partitioned otherwise, e.g. by batch. Additionally, only the manic subtype shares a high degree of pleiotropy with schizophrenia [17]. Aside from psychiatric traits, heterogeneity of genomic associations between known subtypes has been observed in diseases such as lupus [18], multiple sclerosis [19], epilepsy [20], encephalopathy [21], and juvenile idiopathic arthritis [22]. Elucidating the nature of heterogeneity in these traits may also play a

role in addressing the missing heritability problem in GWAS, as hidden heterogeneity reduces power to detect SNP associations [4].

Heterogeneity in disease etiology has also become a concern for clinical applications, as the predictive accuracy of polygenic risk scores is known to vary across different demographics of patients. As most genomic studies to date have been conducted on primarily Northern European populations, accuracy of the discovered predictors, measured as R-squared, is lower in other populations, raising the possibility of inequities in care by the direct application of these PRSs [23]. Even if these concerns are mitigated by future large studies conducted in underserved populations, recent work has shown that PRS accuracy further varies across other covariates such as age and sex [24]. Therefore, methods to develop population-differentiated PRSs and detect deficiencies in existing PRSs are urgently needed before predictive genomics can be widely integrated into precision medicine.

To date there have been few strategies to identify subtypes in GWAS cohorts, largely due to two challenges: the very small signals typically found in polygenic traits, and the presence of confounding sources of heterogeneity such as batch effects. One method [25] purports to discover strong evidence of subtyping in schizophrenia by non-negative matrix factorization of the cohort genotype data, interpreting the hidden factors as different subtypes. However, this work failed to take into account alternative sources of heterogeneity, such as population stratification and linkage disequilibrium, that might produce spurious results [26, 27]. Another method, reverse GWAS [28], applies a Bayesian latent factor model to partition SNP effect sizes and individual membership into a set of latent subtypes so that the likelihood of phenotype predictions within each subtype is maximized. The method is reported to detect subtypes that may be suggestive of clinical implications, such as a possible differential effect of statins on blood glucose levels. However, this approach is under-powered to detect heterogeneity in single phenotypes, and thus is intended for simultaneous predictions across multiple observed phenotypes. Additionally, many of the reported phenotypes are quantitative, which allows for more accurate estimation of effect sizes, and thus more accurate subtyping, than in case/control phenotypes. Therefore methods of this flavor may struggle to detect subtypes among single case/control phenotypes, in which the quantitative liability score is hidden.

Within-phenotype heterogeneity has also surfaced as a possible confounding factor in the discovery of pleiotropic associations between phenotypes [29]. Assuming a GWAS model of disease risk, ideal pleiotropy would involve a single variant significantly associated with two observed phenotypes, producing a genomic correlation between those phenotypes. However, the presence of distinct subtypes in one or both phenotypes may alter the conclusions derived from pleiotropic analysis. For example, two additional subtypes of depression have been characterized by either episodic or persistent experiences of low mood. Of the two, the persistent subtype is more closely associated with childhood maltreatment, and only in persistent cases is an association found between childhood maltreatment and a particular variant of the serotonin transporter gene [30, 31]. Misclassification is another possible source of heterogeneity leading to spurious pleiotropic relationships between phenotypes. For example, a significant percentage of patients diagnosed with either bipolar disorder or schizophrenia have their diagnoses later corrected to reflect the other disease [32]. As bipolar disease and schizophrenia are understood to be highly pleiotropic [5, 33, 34], these misclassifications have the potential to skew analyses of genetic correlation between the two phenotypes.

Recent work by Han et al. [35] has sought to address the detection of heterogeneity specifically in the context of pleiotropic phenotypes. The proposed method, BUHMBOX, operates on a matrix comprising cases for one disease genotyped over the associated SNPs for a second disease. When only a subset of cases are also cases for a second disease, individuals within that subset will exhibit a slightly higher ascertainment for the risk alleles included in the matrix. In

a non-heterogeneous pleiotropic scenario, these risk alleles would instead be randomly distributed among all included individuals rather than co-occurring in a subset. When multiple risk alleles are overrepresented in a subset, they are positively correlated across all individuals in the matrix, and these positive correlations serve as evidence of heterogeneity.

We propose a generalized method called CLiP (Correlation of Liability Predictors) that leverages these correlations more broadly to detect heterogeneity in single-trait GWAS, rather than strictly in two labeled pleiotropic traits. In comparison to BUHMBOX, CLiP improves overall power to detect heterogeneity while remaining robust to false positive confounding factors such as ancestry. CLiP also detects heterogeneity arising from multiple subtypes with highly dissimilar PRSs, which produce specific correlation patterns that attenuate positive heterogeneity signals. These benefits, however, are contingent on the assumption that case/control GWAS behaves according to a liability threshold model rather than a logistic model. Although these models are commonly interpreted to be interchangeable [9], they produce differing SNP-SNP correlations among cases, resulting in different heterogeneity scores.

The goals of this work are fourfold: First, we demonstrate that in a homogeneous (null) set of cases in a case/control cohort, predictors with effect sizes of the same sign are not uncorrelated as stated by Han et al. [35] but negatively correlated, and are expected to produce negative heterogeneity scores. However, the magnitude of the negative bias differs significantly depending on whether the logistic or liability threshold model is assumed for polygenic traits. Second, we evaluate the power of CLiP across realistic GWAS scenarios, and demonstrate its utility by identifying heterogeneity in schizophrenia. Third, we develop an extension of CLiP to accommodate parameters that are not binomial genotypes, but rather continuous predictors such as expression data, which we term CLiP-X. Finally, we further extend CLiP to identify heterogeneous subgroups in quantitative phenotypes, where no clear delineation between cases and controls exists, by weighting correlations according to polygenic risk scores, which we term CLiP-Y.

## Results

While the Introduction described previous methods to partition heterogeneous SNP effects or cohorts into distinct clusters, the highly polygenic nature of most phenotypes renders these methods largely under-powered for single trait GWAS even when data sizes are large. CLiP serves as a compromise on this task by providing a well-powered method to detect potentially disease-relevant heterogeneity, without further decomposing detected signals into clusters. In this regard CLiP can serve as an initial test to flag heterogeneous data sets for further study. CLiP increases power by aggregating pairwise correlations between disease-associated SNPs from summary statistics. Fig 1 depicts genotype and SNP correlation matrices for cases in homogeneous and heterogeneous scenarios. These scenarios are described further in the Methods section.

### SNP-SNP correlations differ in logistic and liability threshold models

Both the logistic and liability threshold models are generalized linear models that transform a linear predictor (the log-odds or the liability score) into a probability of assigning a binary label. The models share highly similar inverse link functions (a sigmoid versus a normal CDF function), and they are understood to produce regression coefficients that can be directly transformed from one model to another by an invertible function (see Methods). However, we found in practice that case/control genotype data simulated from a logistic model and its conversion into a liability threshold model do not produce identical correlation patterns. To demonstrate this, we simulated 10 associated SNPs with allele frequency

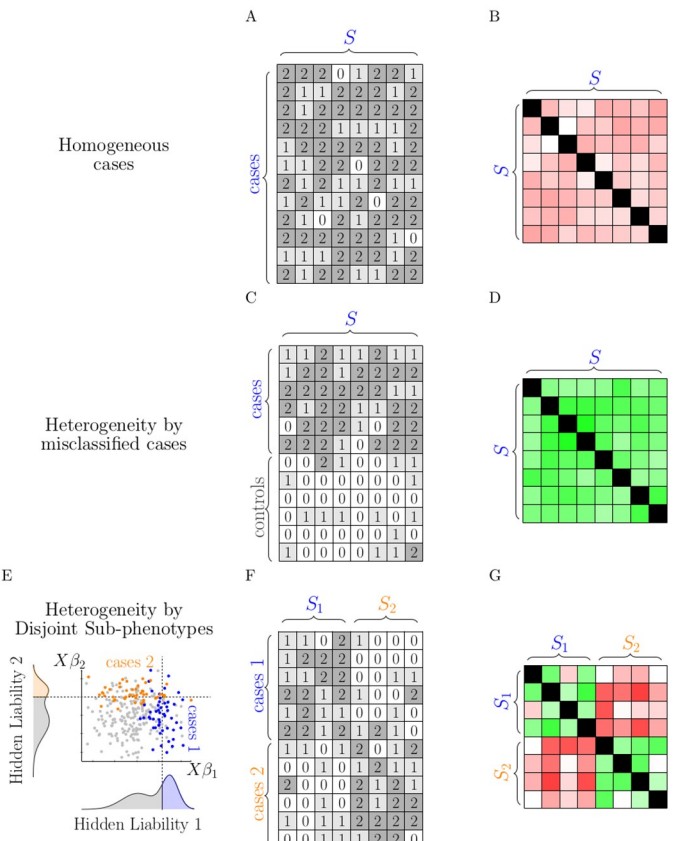

**Fig 1. Depictions of genotype matrices (A,C,F) and SNP correlation matrices (B,D,G) expected of homogeneous and heterogeneous case cohorts.** For homogeneous cases **(A,B)**, SNPs are uniformly ascertained, but negative correlations exist between any pair of associated SNPs. For heterogeneous cases comprising a mixture of true cases and misclassified controls **(C,D)**, SNPs are ascertained in a subset of individuals, creating positive correlations between SNPs. For heterogeneous cases comprising disjoint sub-phenotypes **(E,F,G)**, associated SNP subsets $S_1$ and $S_2$ pertain to two independent PRSs, and passing the threshold of at least one of these PRSs is sufficient to select a case **(E)**. Genotypes sampled from this model produce a mixture of positive and negative correlations.

fixed to 0.5 and a disease prevalence of 0.01. The results of these tests are shown in Fig 2 as a function of increasing odds ratio. At low odds ratios expected of GWAS, genotypes generated from a logistic model produce a largely equivalent ascertainment for risk alleles as its converted liability threshold model, as measured by the mean difference in effect allele count between cases and controls (Fig 2A). Likewise, sample prevalences of randomly sampled population controls from both methods are largely equivalent at low odds ratios (S1 Text and S1 Fig). But over the same simulations the heterogeneity scores, produced by summing pairwise correlations between associated SNPs (see Methods), become significantly more negative when simulating from a liability threshold model than from a logistic model. The expected heterogeneity scores of homogeneous sets of cases generated from logistic and liability threshold models are shown in Fig 2B. Expected scores in the logistic model begin to deviate significantly from zero at an odds ratio of 1.3, larger than SNP effects typically seen in polygenic traits. The sensitivity and specificity of heterogeneity detection by correlations is therefore dependent on the selection of regression model. To further understand the difference between these two models, we compared the standard normal liability distribution against its equivalent in the logistic model, the derivative of the sigmoid function. These

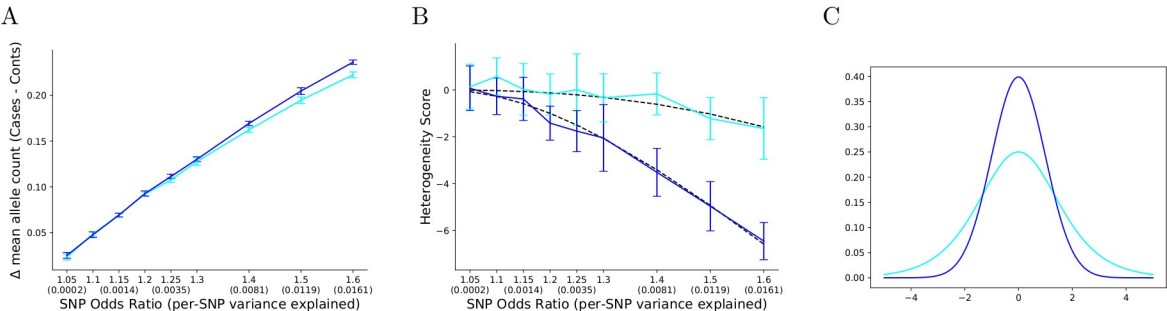

**Fig 2. Diverging correlation behavior of homogeneous cases generated from logistic and liability threshold models.** In all panels, performance of cases generated from a logistic model are shown in cyan, whereas cases generated from a liability threshold model with equivalent properties and converted effect sizes is shown in blue. **(A)** The mean difference in allele count between homogeneous cases and controls in simulated cohorts from logistic models and their conversions to liability threshold models. For each odds ratio, the equivalent per-SNP variance explained on the liability scale is shown in parentheses. Homogeneous cases were generated with a prior prevalence of 0.01 and 10 SNPs with allele frequency 0.5 and odds ratio specified on the x-axis. The mean and standard deviation of 10 trials is plotted for each parameter set. For most of the range of odds ratios expected in GWAS, these models behave identically. **(B)** Heterogeneity scores of the same simulated cohorts in Panel A, with blue denoting the liability threshold model and cyan the logistic model. For each model, a black dotted line denotes the expected value of the score based on effect sizes and allele frequencies. While both models produce negative scores due to negative correlations between SNPs in cases, the bias is notably less negative in the logistic model, and within the range of values typical of complex traits, can be assumed to be close to 0. **(C)** The liability (normal) distribution (blue) and the derivative of the sigmoid function (cyan).

distributions are shown in Fig 2C, in which we observe that the sigmoid derivative has a larger variance than the standard normal distribution.

To contrast the performance of the logistic model with the Risch model, in which SNPs are uncorrelated (see Methods), we simulated case/control cohorts from both models using the same odds ratios, cohort sizes, and prevalence of 0.01. Heterogeneity scores from these simulated cohorts are shown in S2 Fig under large odds ratios of 1.2 (panel A) and smaller odds ratios of 1.06 (panel B). Although both models are multiplicative over odds ratios, the resulting SNP-SNP correlations among their respective cases differs. The Risch model guarantees independence between SNPs in cases due to disease risk being a simple product of odds ratios, so homogeneous cases generated from this model produce heterogeneity scores of 0 regardless of odds ratio magnitude, and heterogeneous cases produce highly positive scores. When generated by a logistic model, however, homogeneous cases become significantly negative due to SNP-SNP anti-correlations, and positive scores of heterogeneous cases are similarly attenuated. But this behavior only occurs when odds ratios are large, and for the majority of highly polygenic traits, the behavior of heterogeneity scores does not differ appreciably from the Risch model, as shown in panel B of S2 Fig.

## CLiP: Correction for negative correlation bias in the liability threshold model

To demonstrate the effects of correlated predictors on heterogeneity detection in the liability threshold model, we evaluated CLiP and BUHMBOX scores on simulated homogeneous and heterogeneous cohorts. Simulation parameters were set to approximate those reported for schizophrenia in Ripke et al. [36]: genotypes over 100 associated SNPs were sampled according to a fixed risk-allele frequency of $p = 0.2$. Effect sizes were set to a fixed value to produce a desired variance explained of 0.034 in a standard normal PRS distribution. SNPs of control cohorts were sampled independently according to population allele frequencies. Homogeneous case sets were generated by repeatedly sampling controls and selecting individuals whose PRSs pass a threshold corresponding to a prevalence of 0.01. Heterogeneous cohorts

A

B

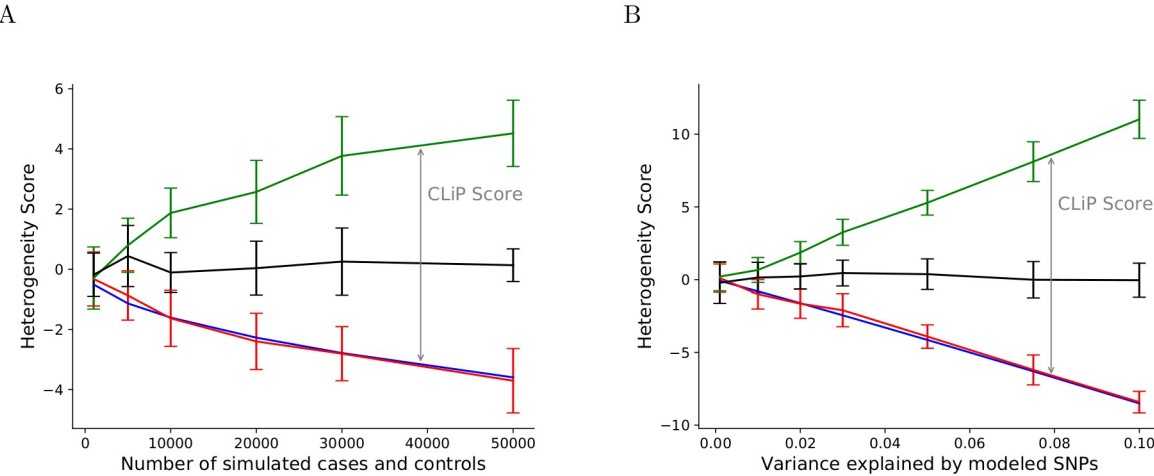

**Fig 3. CLiP performance on simulated control, homogeneous, and heterogeneous cohorts generated with a liability threshold model.**
**(A)** Heterogeneity scores (y-axis) on simulated case/control cohorts as a function of sample size (x-axis). Simulations are run with a PRS of 100 SNPs with total variance explained of 0.034. Heterogeneous cohorts (Green) are equal-proportion mixtures of controls (Black) and homogeneous cases (Red). The expected homogeneous score (Blue) is calculated from effect sizes and allele frequencies of PRS SNPs only, and should be used as the true null score in CLiP. **(B)** Heterogeneity scores (y-axis) as a function of variance explained (x-axis) with a fixed sample size of 30,000 cases and 30,000 controls. For each panel, the mean and standard deviation of 20 trials is plotted.

were created by combining an equal number of homogeneous cases and controls. The scores of these cohorts were evaluated over a range of sample sizes keeping variance explained constant at 0.034 (Fig 3A), and a range of total variance explained values keeping the sample size constant at 30,000 cases and 30,000 controls (Fig 3B). Regardless of both parameters, heterogeneity scores of control populations with independently sampled SNPs (black) follow a standard normal distribution centered at 0. Heterogeneous cohorts (green) exhibit ascertainment of PRS SNPs in one subset of individuals and not the other, resulting in positive correlations between those SNPs taken over all individuals. The weighted sum of these correlations then produces positive heterogeneity scores, which increase when signals increase either by increasing the sample cohort size or the SNP variance explained. Lastly, as these cohorts are simulated from a liability threshold model, homogeneous cases (red) produce negative scores due to competing ascertainment between SNPs as described in S1 Text, and this negative bias likewise increases with increasing signal. Given only knowledge of the SNP effect sizes, effect allele frequencies, and number of cases and controls, we can accurately predict the expected score of homogeneous cohorts (blue) at any cohort size and variance explained.

Power calculations for these results are shown in S1 Table. In Fig 3, we define a positive result to be a true heterogeneous cohort passing a one-sided standard normal confidence interval threshold of 95% given the null CLiP score of a liability threshold homogeneous case set. We compare p-values assuming the liability threshold null given by CLiP and assuming a null value of zero in conventional BUHMBOX. It is clear that correcting for the liability threshold null improves power: for a fixed variance explained of 0.034, CLiP achieves a 99% sensitivity among 20 trials with only 10,000 cases, whereas BUHMBOX requires 30,000 cases to achieve a 100% sensitivity. Additionally, we tested the performance of CLiP with respect to the fraction of individuals in the case mixture that are true cases, shown in S3 Fig and S2 Table. Predictably, we found that maximum power to detect heterogeneity was achieved with an even split between true and misclassified cases, maximizing the entropy. A high percentage representation of true cases decreases the score to negative values, converging to that

of homogeneous case sets, whereas a high percentage representation of controls decreases the score to 0.

Lastly, CLiP is robust to confounding factors generating heterogeneity, provided that these factors exhibit the same patterns in both cases and controls. In particular, ancestry is a source of heterogeneity which must be accounted for in all cohorts. To demonstrate that CLiP is robust to background heterogeneity, we simulated case/control cohorts assuming that the population comprises two ancestry subgroups. The set of simulated SNPs is subdivided so that within each ancestry subgroup alternating halves of the SNPs are assigned effect allele frequencies of $0.5 + p'$ and $0.5 - p'$, with the value of $p'$ set so that the $F_{st}$ among controls is a specified value. Controls are sampled equally from the two subgroups, but cases are selected by thresholding sampled sets of these controls and so are not guaranteed to contain equal subgroup representation. Performance of CLiP in these simulations is shown in S4 Fig. We find that increasing the value of $F_{st}$ attenuates the magnitudes of both homogeneous and heterogeneous case scores toward zero. While power is largely unaffected, the greater concern is specificity, as the presence of these two subgroups creates an alternate heterogeneity signal from disease-related subtyping. In panel B of S4 Fig, we find that specificity over 20 trials remains at 0.8 for an $F_{st}$ of 0.05, with the performance of trials with smaller $F_{st}$ values not differing appreciably from the expected homogeneous score. An $F_{st}$ of 0.05 is double that observed between Finnish and Southern Italian populations, the largest value among pair-wise comparisons of European populations [37]. We conclude that CLiP is robust for cohorts sampled within single well-defined populations, for which $F_{st}$ is expected to be far smaller.

## Heterogeneity by distinct subtypes

In contrast to heterogeneity created by subsets of misclassified cases, we also consider heterogeneity arising from multiple potentially independent sub-phenotypes, each with a distinct PRS, such that an individual is considered to be a case when it is a case for one or more of these sub-phenotypes. Discovering heterogeneity in these cohorts is more challenging because correlations between SNPs involved in different sub-phenotype PRSs may be negative as depicted in Fig 1G, reducing the heterogeneity score when summed. Additionally, while we depict misclassification and subtyping as two distinct sources of heterogeneity, most phenotypes and case/control cohorts will fall between these two extremes as PRSs exhibit different variances explained with respect to different subtypes. We can interpret misclassification as a particular form of subtyping in which one subgroup exhibits a very small variance explained with respect to known SNP associations, as can occur when transferring PRSs between different populations or age or sex cohorts [24]. We have modeled the spectrum between misclassification and subtyping in Fig 4A. Assuming there exist two subtypes within a case set with associated SNP $S_1$ and $S_2$ and effect sizes $\beta_1$ and $\beta_2$ fixed across all SNPs, we keep $\beta_1$ fixed to a positive value while varying the magnitude of $\beta_2$. When $\beta_2$ is 0, the risk alleles of set $S_2$ are not ascertained for any subgroup of cases, resulting in small weight factors $w$ and a negligible contribution to the CLiP score. The remaining SNPs $S_1$ then reduce to the misclassification scenario. As the magnitude of $\beta_2$ increases, the subtyping pattern approaches that depicted in Fig 1F.

We find in Fig 4B that both subtyping and misclassification produce score significantly larger than that expected of a homogeneous case set in a liability threshold model, but also that the raw score prior to correcting by the null expectation differs between the misclassification and subtyping scenarios. Misclassification ($\beta_2 \to 0$) tends to produce highly positive scores, as all ascertained SNPs are ascertained among one subgroup, and not ascertained in the other,

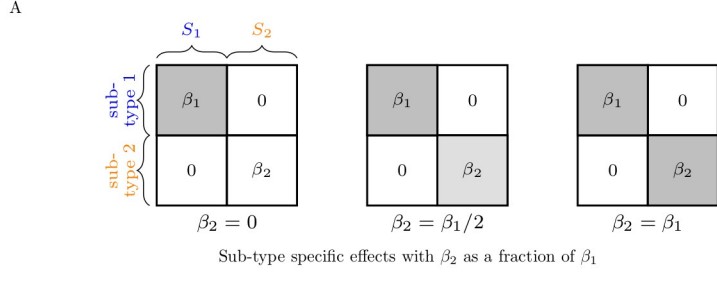

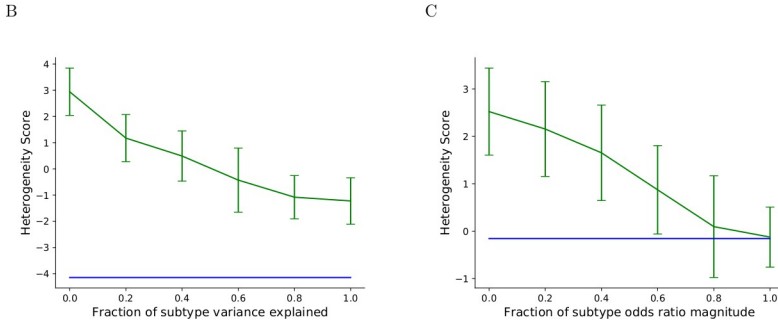

**Fig 4. Performance of heterogeneous scores on case sets heterogeneous by misclassification and by subtype.** While misclassification and subtyping represent two alternate etiologies for observed heterogeneity, most real case sets likely fall between these two extremes. **(A)** Given two subtypes $S_1$ and $S_2$ with fixed subtype-specific SNP effect sizes $\beta_1$ and $\beta_2$, as well as cases belonging to each, a spectrum can be drawn over increasing variance explained of one of the subtypes. When $\beta_2$ is zero, the SNPs in $S_2$ do not contribute to the PRS of any subset of cases and so the heterogeneity score is driven by the set $S_1$, mimicking the misclassification scenario. As $\beta_2$ increases in magnitude, two distinct subtypes are formed. **(B)** Heterogeneity scores (green) of cohorts simulated from a liability threshold model according to the subtype pattern depicted in A, with equal proportion of cases generated from each subtype. The x-axis represents the liability variance explained by SNPs in $S_2$ as a fraction of the variance explained by SNPs in $S_1$. The blue line depicts the expected score of homogeneous cases under the liability threshold model, absent distinct subtypes. All scores of the simulated heterogeneous cohorts are significantly larger than the null value, indicating heterogeneity is detectable at both extremes, but the raw value of the score in cases is highly positive under the misclassification scenario and slightly negative under subtyping. **(C)** The same experiment in B but generated from a logistic model with $S_1$ odds ratio set to 1.06, and $S_2$ odds ratio set to a fraction of that value denoted on the x-axis. Without the high negative correlation bias present in the liability threshold model, the subtyping scenario under the logistic model produces heterogeneity scores near 0 and so cannot be detected.

producing identical correlation patterns between all SNP pairs. In the subtyping scenario ($\beta_2 \to \beta_1$), SNP pairs within either $S_1$ or $S_2$ are positively correlated, but SNP pairs with one $S_1$ SNP and one $S_2$ SNP are negatively correlated, significantly attenuating the score. This pattern is also observed in the logistic model in Fig 4C, but simulations approaching the subtyping extreme have heterogeneity scores which converge to the null score near zero, indicating that under a logistic model, subtyping is not detected as heterogeneity. The subtyping scores, however, do not converge to zero in the liability threshold model but to a negative value. As SNPs in both $S_1$ and $S_2$ are slightly ascertained over the entire case set, a negative correlation bias in the liability model is still observed in these SNPs, decreasing the score from zero.

Lastly, we tested the performance of CLiP in the subtyping scenario when associated SNPs are subdivided amongst an increasing number of hidden subtypes. We tested the performance of CLiP by fixing the number of cases and controls at 50,000 each, the total number of SNPs at 100, and the total variance explained at 0.05, while varying the number of sub-phenotypes and the fraction of SNPs that are shared across all sub-phenotypes. When this fraction is zero, the sub-phenotypes are completely independent, and the SNPs are divided into mutually exclusive subsets associated with each sub-phenotype. When the fraction is non-zero, that fraction of

SNPs has the same effect size across all sub-phenotypes. Results of these simulations are shown in S5 Fig as well as S3 Table. Note that by dividing associated SNPs into associations with particular sub-phenotypes, the total variance explained for each sub-phenotype is reduced, and the observed variance explained of the entire heterogeneous cohort will be lower in a simple linear regression.

## CLiP-Y: Quantitative phenotypes

To demonstrate heterogeneity detection in quantitative phenotypes, we simulated homogeneous cohorts by sampling phenotypes from the true PRS-dependent distribution $Y \sim N(X\beta, 1 - h_{SNP}^2)$, and we simulated heterogeneous cohorts by combining equal-sized samples from the true distribution and from a PRS-independent standard normal distribution. In lieu of scoring the difference between case and control correlations, CLiP-Y scores a weighted correlation with phenotype-dependent function $\phi(Y)$ against a conventional unweighted correlation over the same individuals.

In practice, we found evaluating a given choice of $\phi$ over phenotypes converted to percentiles improved performance for all learned weight functions, possibly because percentiles limit the domain of the phenotype over which the weight function must be positive, reducing the contribution to the score calculation by extreme PRS values. In addition to testing preselected functions for $\phi$, we performed a local search over polynomials of increasing degree (see Methods), finding the optimal polynomial functions shown in Fig 5A. All polynomial functions converged to highly similar concave functions. This is due to the balancing effect of the normalization factor on the sum of correlations: while correlations of PRSs at the high end of the distribution are more extreme because these individuals more closely resemble "cases," a high weight value at the higher end of the PRS spectrum means that the normalization factor also shrinks the magnitude of the score. To demonstrate that optimal weight functions are concave functions over the range of PRS percentiles, we tested weight functions that sum up two indicator functions scanning across the range of percentiles in [0, 1], one increasing, for an interval ending at 1, and another decreasing, for an interval ending at 0, and evaluated heterogeneity detection performance as shown in S6 Fig. The best performing functions are those where the increasing function threshold is near but not at 0, and the decreasing function threshold is near but not at 1, producing a function similar to the concave polynomials found in Fig 5A.

In the absence of a method for scoring continuous phenotypes, a naive approach using conventional case/control heterogeneity scoring would involve setting an arbitrary threshold in the distribution of phenotypes by which to partition the cohort from a continuous phenotype into cases and controls. This is equivalent to applying a step weight function over the phenotypes. We compare our continuous heterogeneity test to thresholds at various phenotype percentiles. CLiP-Y scores using a step weight function are calculated on simulated data over varying sample sizes and values of variance explained, shown in S7 and S8 Figs. Both the continuous heterogeneity test and the arbitrary threshold tests are standard normally distributed in the null scenario, when no heterogeneity is present.

We evaluated the performance of our polynomial weight functions against the step function and other weight functions by the difference between the CLiP-Y score of simulated heterogeneous cohorts and the calculated expected homogeneous score. A derivation for the expected score is shown in S1 Text, and comparisons with randomly sampled cohorts are shown in panel A of S9 Fig and panel A of S10 Fig. The mean scores over 20 trials conducted in 100,000 cases and 100,000 controls are shown in Fig 5B, as a function of variance of the quantitative trait explained by SNPs. We observe that the sigmoid function, a smoothed step function,

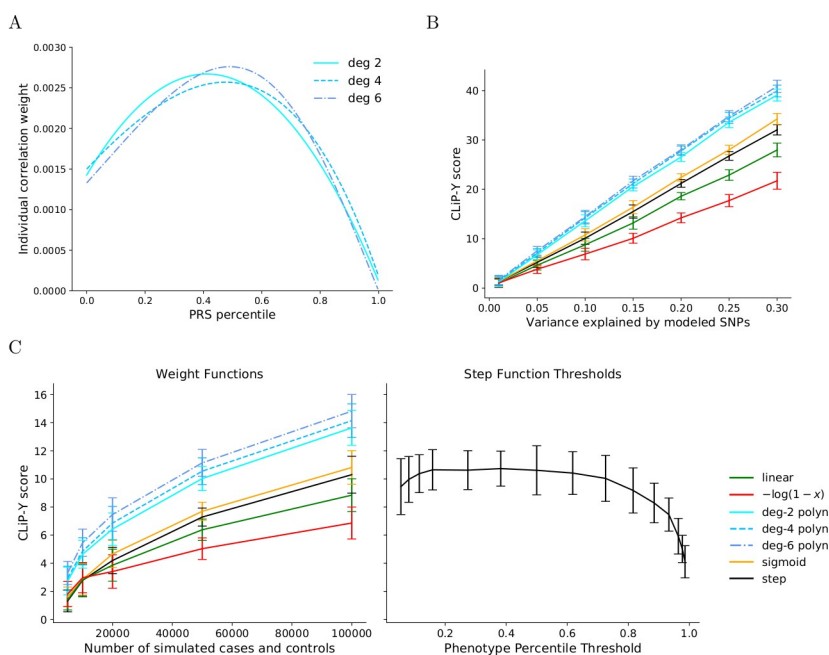

**Fig 5. CLiP-Y performance on simulated cohorts with a quantitative phenotype. (A)** Learned weight functions $\phi(x)$ for scoring heterogeneity in quantitative phenotypes. A local search over polynomial coefficients is performed such that the resulting function maximizes the difference between the CLiP-Y scores of simulated samples of homogeneous and heterogeneous cohorts. These functions are fixed for all subsequent tests. **(B)** Tests for heterogeneity in quantitative phenotypes using multiple weighting functions over individuals, including those in panel A, as a function of variance explained by the PRS. Plotted are mean scores of simulated heterogeneous cohorts over 20 trials minus the expected score of a homogeneous cohort. One Hundred SNPs are simulated with cohorts of 100,000 cases and 100,000 controls. **(C)** The left panel depicts the same simulations in B, but over varying sample size with a fixed variance explained of 0.1. For comparison these scores are plotted on the same Y-axis as scores generated from step function weights at various thresholds on the percentile scale of a standard normal quantitative phenotype distribution. For each of these step function scores, the expected homogeneous score is estimated by the mean of 20 sampled homogeneous cohorts, to limit computation time.

slightly outperforms the step function, whereas a linear function and $-\log(1-x)$, intended to assign large weights to the positive extreme of the phenotype distribution, both underperform. The three learned polynomial functions perform similarly and significantly outperform all manually selected functions.

The same relative performance of the weight functions is observed when tested against an increasing cohort size with a fixed variance explained of 0.1, shown in the left plot of Fig 5C. For comparison, the right plot shows CLiP-Y scores on the same scale for step functions with thresholds set at different percentiles of the phenotype distribution and simulated in cohorts of 100,000 cases and 100,000 controls. Note that as expected, the best performing percentile threshold is some intermediate rather than extreme value.

## CLiP-X: Quantitative predictors

CLiP-X performs heterogeneity detection in case/control liability threshold models with quantitative predictors. While the identity of these predictors may be arbitrary, we consider in particular the application of CLiP-X to Transcriptome-Wide Association Studies (TWAS), in which imputed gene expression variables are tested for association with disease status [38, 39]. In a cohort with observed genotypes $G$ and gene expression $Z$, TWAS performs association tests to estimate SNP-gene effect sizes $\beta$. In a larger genotype cohort $G'$, gene expression is

then imputed by $Z' = G'\beta$ and used to estimate gene-trait effect sizes $\alpha$. The effect sizes $\alpha$ are interpreted as the fraction of SNP variance mediated by the involved genes. CLiP-X tests for heterogeneity among these quantitative gene-trait associations. In our simulations, we allow for correlations between genes due to shared SNP effects. To model this, we generate gene expression variables as linear functions over a common set of SNPs along with normally distributed error, with a specified SNP-gene variance explained.

To demonstrate the performance of CLiP-X in case/control cohorts with continuous predictors instead of binomial SNPs, we simulated quantitative expression data by first sampling 100 SNPs, and then sampling 10 normally-distributed expression variables as linear functions of these SNPs with variance explained $V_G^2$ set to 0.1 for each gene (see Methods). These 10 genes determine simulated individuals' case/control disease status according to a liability threshold model with variance explained $V_E^2$. This simulation scenario mimics a transcriptome-wide association study, but we additionally allow for the inclusion of a high frequency of trans-effects by generating all simulated genes from the same 100 SNPs, and correct for the resulting confounding correlations between genes. We ran 20 trials across a range of sample sizes and total trait variance explained by expression $V_E^2$, to evaluate the performance of the continuous variable test statistic in true heterogeneous cases, homogeneous cases, and independently sampled controls. The results are presented in Fig 6 and S11 Fig.

We observe that for all sample sizes, the heterogeneity score is approximately distributed with mean 0 and standard deviation 1 in control cohorts. As predicted, the homogeneous case group exhibits highly negative correlations between associated SNPs, and the resulting CLiP-X score can be accurately estimated from expected correlations (blue) using knowledge of summary statistics only. Additionally, because the input predictors of CLiP-X are normally distributed, we can confirm by Fisher's transformation that sample correlations between predictors exhibit the expected variance around the calculated null score, indicating the null score accurately estimates the true expected correlation (S12 Fig). This estimate should serve as the null when evaluating GWAS cohorts in practice, when a truly homogeneous cohort is not available. By comparison to this true null, many more heterogeneous cohorts are detectable which would not have passed a significance threshold with the null centered at 0, especially those with sample sizes of less than 10,000 cases.

## Application to schizophrenia and neuroticism

To evaluate the performance of CLiP on real genomic data sets, we tested for the presence of heterogeneity in case/control cohorts collected by the Psychiatric Genomics Consortium (PGC) for schizophrenia, from which 108 associated loci were discovered by GWAS [36]. After transforming these known effect sizes to the liability scale and excluding 8 SNPs from further study (see Methods), the total variance explained by the remaining 100 genome-wide significant SNPs was approximately 0.027, suitably close to the 0.034 SNP variance explained reported in Ripke et al. [36] when accounting for removed SNPs. We calculated heterogeneity scores for cases and controls over individual cohorts, shown in Fig 7, as well as meta-analysis scores over all cohorts as described in the Methods, shown in Table 1. Generally, we observe more positive heterogeneity scores for larger cohorts, though few pass a significance p-value threshold of 0.05. The scores in Table 1 are organized by ascending p-value, and a Benjamini-Hochberg procedure is conducted with a false-discovery rate of $\frac{1}{3}$. Cohorts with p-values lower than the critical values determined by this FDR are separated by a dashed line. On an individual basis the vast majority of these cohorts are too small to be conclusively tested for heterogeneity, as the sample variances of correlations between SNPs is high. By performing a single test over all cases and controls combined, we obtain a significant p-value of $8.54 \times 10^{-4}$, though

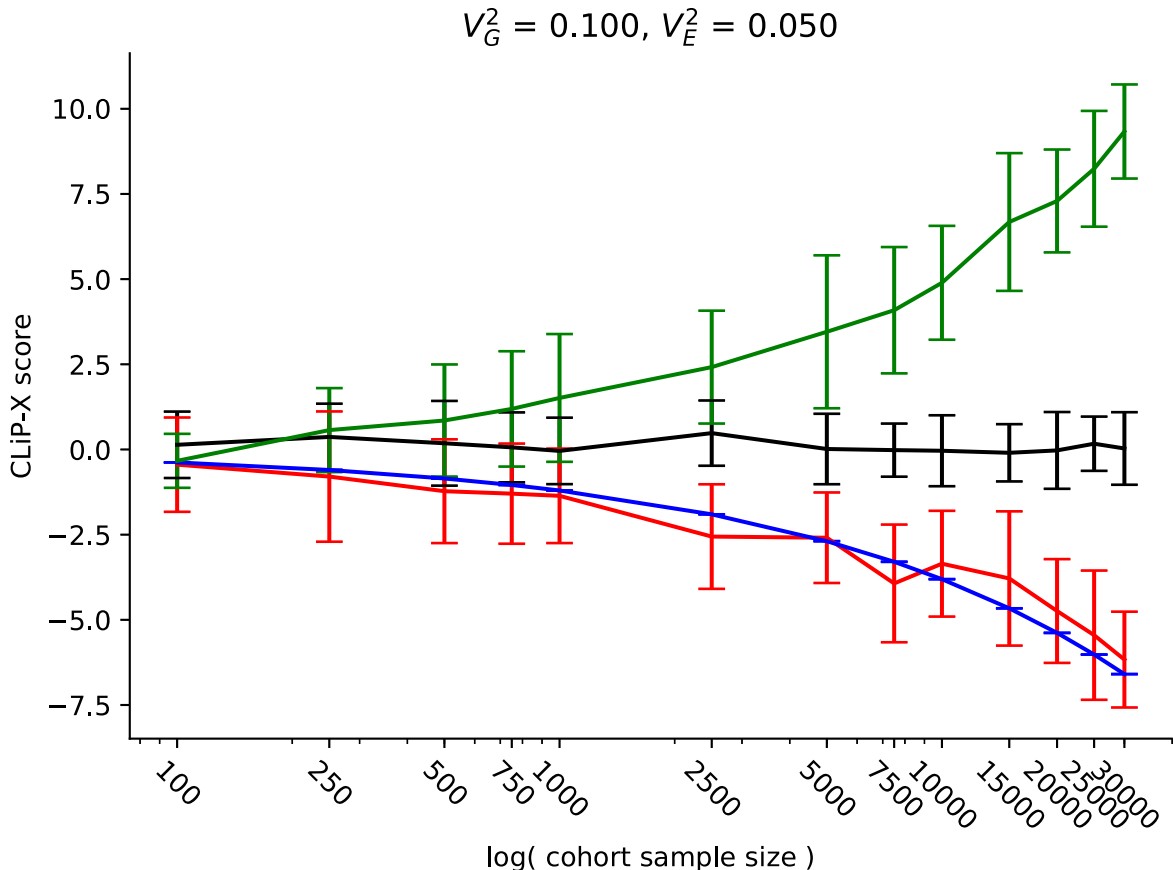

**Fig 6. CLiP-X performance on simulated control, homogeneous, and heterogeneous cohorts with quantitative predictors.** Controls (black) have no criteria for selection placed on their generated quantitative predictors; homogeneous cases (red) are selected according to a liability threshold over predictors; and heterogeneous cases (green) are an even combination of controls and homogeneous cases. The blue line indicates expected mean scores of homogeneous cohorts calculated from summary statistics of the quantitative predictors. As with discrete SNPs, quantitative predictors are negatively correlated among homogeneous cases.

some heterogeneity may be contributed by batch effects. By summing scores across cohorts, we obtain a larger but still significant p-value of 0.011, suggesting that while batch effects contribute to detected heterogeneity, they do not completely account for all heterogeneity observed in the data. Lastly, by applying meta-analysis methods over individual cohort scores, we obtain a Fisher's $\chi^2$ p-value of 0.030, and a Z-score of 2.03, also supporting the presence of a significant heterogeneity signal.

To investigate the potential contribution of demographics to observed heterogeneity, we inferred ancestry subgroups of the largest four case/control cohorts in the PGC data set (clm2, clo3, s234, and boco) and scored cases and controls belonging to each subgroup individually. Ancestry inference was performed using fastStructure [40] with the total number of subgroups set to 2, 3, and 4. Generally, we found that the heterogeneity scores of these subgroups were not consistently smaller than the score of the entire cohorts, suggesting that the source of observed heterogeneity in these cohorts is not attributable to ancestry. The results of these tests plotted against the size of the subgroups and whole cohorts are shown in S13 Fig. Unfortunately, the same analysis could not be performed for sex as among the four largest cohorts, sex information was unavailable for the cohort s234 and significantly imbalanced toward males in clm2.

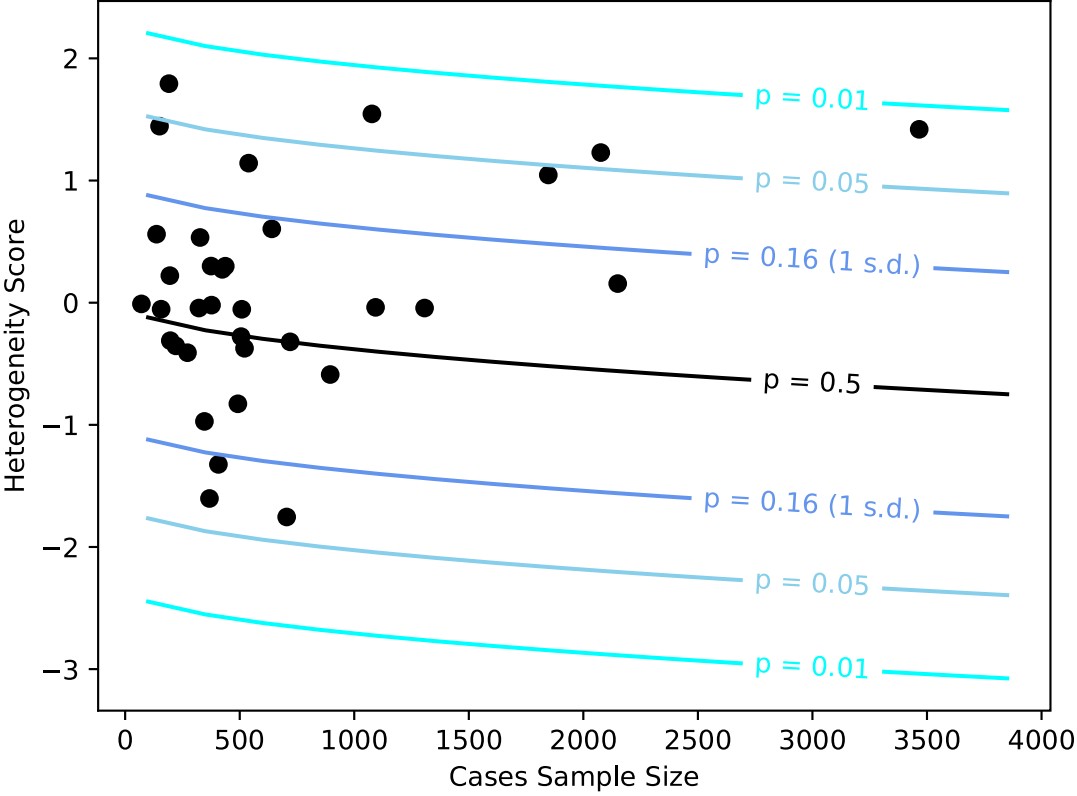

**Fig 7. CLiP heterogeneity scores evaluated over single cohorts in the PGC schizophrenia data set, plotted by the number of genotyped cases.** The black line denotes the expected score given summary statistics reported for schizophrenia and sample sizes specific to each cohort, and colored lines denote z-score thresholds corresponding to particular p-values of significance.

We also applied CLiP-Y to test for heterogeneity in neuroticism in a sample of 10719 individuals from the UK Biobank [41, 42]. Participants are predominantly of Northern European ancestry residing in the United Kingdom. Neuroticism was scored by the UK Biobank according to the revised short form of the Eysenck personality questionnaire [43]. The quantitative phenotype to be evaluated for heterogeneity is the number of yes responses in the 12 questions of the form. Heterogeneity was evaluated in CLiP-Y using all learned polynomial weight functions over the neuroticism score distribution, along with the 0.5 percentile step function representing a balanced pseudo-case/control split. We also tested subsets of the cohort by sex and by known neuroticism subcluster to investigate whether these partitions were responsible for any observed heterogeneity signal. The known subclusters of neuroticism are each obtained by scoring a subset of 4 out of 12 questions on the Esenck short form. These are reported in [44] along with their relevant survey questions to be "Depression" ("lonely", "miserable", "mood up/down", and "fed-up") and "Worry" ("worrier", "nerves", "nervous person", "tense/high-strung"). Individuals were included in these subclusters if they answered "yes" for at least 3 out of 4 questions, and their full 12-question sum-scores were evaluated as before. Results from these tests are shown in Table 2. As described in the Methods, to save on computation time the expected scores for each test were calculated by averaging the CLiP-Y scores of 400 simulated cohorts with independently sampled SNPs given the input summary statistics. We found that CLiP-Y scores for the entire cohort indicated significant heterogeneity across all tested weight functions, with p-values ranging from $1.68 \times 10^{-9}$ to $8.45 \times 10^{-11}$. As expected, all polynomial functions performed similarly, so

**Table 1. CLiP heterogeneity scores for PGC schizophrenia cohorts and their combination.** Cohorts with an asterisk have had a SNP excluded which has zero variance within either the case or control cohort, which would result in an undefined correlation. P-values of scores using the liability threshold null (CLiP) are shown along with p-values using a null of zero (BUHMBOX) in parentheses. An FDR of $\frac{1}{3}$ was used for Benjamini-Hochberg analysis.

| Cohort | Num Cases/Conts | CLiP Score | Expected Score | p-value | FDR critical val |
|---|---|---|---|---|---|
| clm2 | 3466/4297 | 1.42 | -0.77 | 0.014 (0.078) | 0.010 |
| gras | 1077/1226 | 1.55 | -0.42 | 0.025 (0.061) | 0.020 |
| zhh1* | 191/190 | 1.79 | -0.17 | 0.025 (0.036) | 0.029 |
| s234 | 2076/2341 | 1.23 | -0.57 | 0.036 (0.110) | 0.039 |
| pews | 150/236 | 1.45 | -0.16 | 0.054 (0.074) | 0.049 |
| boco | 1847/2169 | 1.05 | -0.55 | 0.056 (0.148) | 0.059 |
| cou3 | 539/692 | 1.14 | -0.31 | 0.073 (0.127) | 0.069 |
| pewb | 640/1892 | 0.60 | -0.38 | 0.163 (0.273) | 0.078 |
| clo3 | 2150/2083 | 0.16 | -0.57 | 0.233 (0.438) | 0.088 |
| lie2 | 137/268 | 0.56 | -0.17 | 0.234 (0.287) | 0.098 |
| msaf | 327/139 | 0.53 | -0.17 | 0.241 (0.297) | 0.108 |
| umeb | 375/584 | 0.30 | -0.26 | 0.286 (0.382) | 0.118 |
| munc | 437/351 | 0.30 | -0.24 | 0.294 (0.383) | 0.127 |
| caws* | 424/305 | 0.27 | -0.23 | 0.307 (0.392) | 0.137 |
| buls | 195/608 | 0.22 | -0.21 | 0.332 (0.412) | 0.147 |
| swe6 | 1093/1217 | -0.04 | -0.42 | 0.352 (0.515) | 0.157 |
| irwt | 1307/1022 | -0.04 | -0.41 | 0.357 (0.518) | 0.167 |
| top8 | 377/403 | -0.02 | -0.24 | 0.414 (0.508) | 0.176 |
| asrb | 509/310 | -0.05 | -0.24 | 0.426 (0.522) | 0.186 |
| ersw | 322/332 | -0.04 | -0.23 | 0.428 (0.517) | 0.196 |
| lacw | 157/466 | -0.05 | -0.19 | 0.445 (0.521) | 0.206 |
| cims | 71/69 | -0.01 | -0.10 | 0.463 (0.504) | 0.216 |
| aber | 720/699 | -0.32 | -0.33 | 0.497 (0.626) | 0.225 |
| lie5 | 506/387 | -0.28 | -0.25 | 0.510 (0.609) | 0.235 |
| umes | 197/713 | -0.31 | -0.21 | 0.539 (0.622) | 0.245 |
| uclo* | 521/494 | -0.37 | -0.27 | 0.540 (0.646) | 0.255 |
| dubl | 272/860 | -0.41 | -0.25 | 0.562 (0.659) | 0.265 |
| swe1 | 221/214 | -0.35 | -0.18 | 0.567 (0.638) | 0.274 |
| ajsz | 895/1593 | -0.59 | -0.41 | 0.572 (0.722) | 0.284 |
| denm | 492/458 | -0.83 | -0.27 | 0.713 (0.796) | 0.294 |
| port* | 346/216 | -0.97 | -0.20 | 0.781 (0.834) | 0.304 |
| cati | 407/391 | -1.32 | -0.24 | 0.860 (0.907) | 0.314 |
| edin | 368/284 | -1.60 | -0.22 | 0.916 (0.946) | 0.323 |
| ucla | 705/637 | -1.75 | -0.32 | 0.925 (0.960) | 0.333 |
| All | 23517/28146 | 1.20 | -1.93 | 8.54e-4 (0.114) | |

only the degree-6 polynomial was applied to further analyses. After partitioning by sex and neuroticism subcluster, we obtained scores which no longer passed a p-value threshold of 0.01 for heterogeneity: a value of.0165 in males, and.0249 in the "Worry" subcluster. However, from simulation results in Fig 5C, we also know that the magnitude of heterogeneity scores is sensitive to cohort sample size and decreases for small sizes. To account for this scenario, for each sex or subcluster partition we calculated the average heterogeneity score of an equivalently-sized random sample of the entire cohort. The difference between the CLiP-Y scores of the partitions and these random samples is not statistically significant, indicating that the lower scores are indeed due to smaller sample size. Therefore the observed

**Table 2. CLiP-Y heterogeneity scores for neuroticism and sub-clusters in the UK Biobank.** Significant heterogeneity was detected in the cohort across all weight functions. When scoring partitions of the data by sex and by known neuroticism subtype, lower scores are observed, but these scores are not significantly distinct from random samples of the cohort of equivalent size ("subsample mean(std)"), indicating that the observed heterogeneity cannot be primarily attributed to these variables.

| Cohort | Num Indivs | Weight Fcn | CLiP-Y Score | Expected Score | p-value | subsample mean(std) |
|---|---|---|---|---|---|---|
| all | 10719 | poly deg-6 | -1.967 | -7.880 | 1.68E-09 | |
| | 10719 | poly deg-4 | -2.138 | -8.474 | 1.17E-10 | |
| | 10719 | poly deg-2 | -2.155 | -8.542 | 8.45E-11 | |
| | 10719 | step | 0.827 | -5.368 | 2.92E-10 | |
| Males only | 4949 | poly deg-6 | -3.401 | -5.534 | 1.65E-02 | -2.189 (0.955) |
| Females only | 5770 | poly deg-6 | -0.702 | -5.974 | 6.75E-08 | -2.008 (0.961) |
| Depressed subcluster | 3037 | poly deg-6 | -1.440 | -4.884 | 2.86E-04 | -2.552 (1.107) |
| Worry subcluster | 1815 | poly deg-6 | -2.621 | -4.583 | 2.49E-02 | -2.887 (1.242) |

heterogeneity in neuroticism is not primarily driven by these variables, and further investigation is needed to identify these sources of heterogeneity.

## Discussion

We present a general framework for identifying hidden heterogeneity among cases given only genotype and phenotype data for a single disease, while being robust to confounding heterogeneity from sources such as ancestry. We derive modified test statistics to account for non-genotype input variables such as expression data, which may be continuous and exhibit confounding correlations in controls due to shared eQTLs. Additionally, we allow for heterogeneity to be scored in quantitative phenotypes that lack a clear distinction between cases and controls which would facilitate a simple dichotomous contrast of correlation patterns.

In addition to detecting heterogeneity, we describe an informal method to discern between heterogeneity arising from misclassification, in which a significant subset of cases do not exhibit elevated risk with respect to the PRS, and subtyping, in which the PRS comprises hidden sub-PRSs with distinct effect sizes and ascertained individuals. Prior to correction by the control score, misclassification produces highly positive scores, whereas subtyping produces negative scores in the liability threshold model and scores near zero in the logistic model (and are thus undetectable). However, the magnitude of the heterogeneity score itself is highly dependent on many variables including sample size, number of SNPs in the PRS, and the magnitude and distribution of effect sizes, and this precludes us from devising a more formal test for one or the other heterogeneity scenario.

Our analysis of SNP correlations among cases also reveals a distinction between the logistic and liability threshold models which until now has been unreported. Wray and Goddard [9] report that when considering both the estimation of individual risk and the magnitude of effect sizes, the logistic and liability models are interchangeable for practical use despite small discrepancies such as the significance of disease prevalence. But as we show in Fig 2, for the same set of simulations, the effect sizes themselves may be highly similar while correlations between SNPs in cases sampled from each of these models differ widely. The magnitude and sign of these correlations depend on the magnitudes and signs of the effect sizes of the SNP pair (see S1 Text), which may impede their discovery: due to high polygenicity, individual correlations are small, and if effect sizes are a mixture of positive and negative values with mean 0, then the mean correlation over all pairs of SNPs will likewise be 0. But when all SNP effects are set to be positive, the sum of all pair-wise correlations among homogeneous cases does differ

significantly from the standard normal distribution centered at 0. When scoring heterogeneity, do we then adopt the liability threshold model to explain case/control GWAS (CLiP), or do we adopt the logistic regression model (BUHMBOX)? There are reasons to assume that the liability threshold model is a better reflection of the underlying etiology. First, the logistic model is itself a threshold model in the space of log odds ratios, with the overall PRS distribution defined by the derivative of the sigmoid function. However, given the central limit theorem, we would expect the PRS distribution to tend toward normality rather than this derivative (see S1 Text for discussion on the normality of PRSs). Additionally, while correlations between SNP pairs are likely too small to detect with confidence, there has been evidence to suggest that entire PRSs are negatively correlated with rare variants. Bergen et al. [45] report that among cases for schizophrenia, individuals possessing rare copy number variants of large effect tend to have lower PRSs over common alleles, a negative correlation. The authors propose that increased risk from rare CNVs decreases the required risk load of the PRS, and that this implies both CNVs and common alleles participate in a single additive liability threshold model. Martin et al. [46] report the same relationship between rare CNVs and PRSs in Attention-Deficit/Hyperactivity Disorder. As the variance explained of rare variants and entire PRSs is larger, we would expect negative correlations to be easier to detect at current sample sizes. In practice however, no cohort can be expected to be truly homogeneous: sampling differences between cases and controls may produce unequal representation of demographic variables or a difference in residual LD after filtering for uncorrelated loci, resulting in slight heterogeneous signals. For these reasons we recommend interpreting the respective null scores of CLiP and BUHMBOX as lower and upper bounds on the score expected of a homogeneous cohort with the given PRS parameters.

We caution against attributing detected heterogeneity directly to the underlying disease etiology. The source of heterogeneity may instead be due to issues such as batch effects arising from data collection or stratification of the PRS across demographic features such as age or sex. Depending on the degree to which these features are known, they can be ruled out as primary contributors to detected heterogeneity by separately scoring subsets of the data partitioned on these features, though this requires data sets large enough post-partition. Detecting heterogeneity due to demographic variables may be of interest itself in certain applications, such as evaluating the performance of PRSs for scoring disease risk when applied across diverse populations. While CLiP does not positively determine the source of heterogeneity within a data set, it serves as a lightweight method to screen for the presence of heterogeneity and a starting point for further investigation.

Lastly, the real data results presented in this manuscript only consider PRS based on SNPs whose association signals are genome-wide significant. This removes concerns of false positive associations within the PRS. PRS constructions that do include lower-significance SNPs explain more heritability, and are an attractive next challenge for finding heterogeneity signals. Aside for the statistical challenge, this future work would require handling much larger sets of SNPs, and therefore larger matrices of correlations.

We envision CLiP as a method to quickly screen many polygenic risk scores for heterogeneity and flag particular traits for further investigation, either as a way to further elucidate potential subtype-specific etiologies, or to detect unwanted transferability issues across different demographic groups. At the grander scheme of human genetics, generalized testing for heterogeneity paves the way for recovering additional layers of the network of effects that explain traits by interacting genetic and other factors. Going beyond the first-order, linear approximation of these effects holds the promise of better explaining mechanisms beyond identification of their contributing input factors.

## Methods

In a case-control GWAS, a heterogeneous cohort of cases can be interpreted as comprising a mixture of hidden case subtypes, each exhibiting an elevated risks of disease according to a unique polygenic risk score. These subtype-specific PRSs are unobserved and may confound discovery of case-control associations. We define two models for generating genotype matrices of heterogeneous cohorts: First, misclassification, where ideally a subset of individuals are not really cases, but have been rather labeled as such despite being controls. This may occur due to erroneous phenotyping, but it may also suggest distinct disease etiologies, some of which are not ascertained for the PRS of interest. Second, a mixture of multiple unobserved sub-phenotypes which are all captured to some degree by the PRS of interest. An idealized example would involve SNPs of a PRS divided between sub-types such that each SNP was truly associated with only one subtype, and no SNP shared associations with multiple subtypes. A case is observed if the individual passes the liability threshold of at least one of these sub-phenotype PRSs. Fig 1 displays idealized genotype matrices and correlation matrices for each of these models along with the homogeneous null scenario, in which all cases are selected according to the same PRS. The column set $S$ comprises associated SNPs reported in GWAS summary statistics, with the counted allele selected so that the corresponding effect size is positive. As described in Results, associated SNPs participating in the same PRS are negatively correlated over a set of cases selected according to that PRS (panel B). When the cohort comprises both cases and misclassified controls, the pattern of ascertainment of risk-alleles is consistent for particular individuals across all SNPs, resulting in positive correlations between SNPs (panel D). Panel E depicts a mixture scenario with two hidden disjoint PRSs. Individuals labeled as cases of the observed phenotype may be in reality a case for sub-phenotype 1 only (blue), sub-phenotype 2 only (orange), or both, whereas controls are observed as such (grey). The presence of cases for multiple hidden sub-phenotypes produces a mixture of positively and negatively correlated SNPs depending on the membership of the compared SNPs (panel G).

The goal of CLiP is to distinguish a heterogeneous cohort from one that comprises only homogeneous cases and controls for a single PRS. In the following sections, we first describe a correction (CLiP) to current applications of heterogeneity scores [35], where we account for negative correlations expected of case/control data sampled from a liability threshold model. Next we present adaptations of this general method to studies with quantitative predictors such as expression measurements rather than SNPs (CLiP-X), and also with quantitative phenotypes for which there is no strict definition of a "case" (CLiP-Y). Additionally, we describe the generative process for simulations of homogeneous and heterogeneous PRS data used to test the performance of these methods.

### Regression models for case-control data

Here we describe three models characterizing risks of disease based on odds ratios or effects of SNPs. We will demonstrate in the Results section that only in the Risch method is independence of PRS SNPs preserved among cases, whereas both the logistic and liability threshold models introduce correlations between these SNPs after ascertaining for disease status.

**Multiplicative (Risch) model.** The Risch model describes disease risk as a prior disease prevalence multiplied by a product of relative risks corresponding to each risk factor. For $M$ SNPs in a PRS with relative risks $RR_m$ and constant $A$, an individual with genotypes $x$ has the following disease risk:

$$P(y = 1|x) = A\prod_{m=1}^{M} RR_m^{x_m} \tag{1}$$

For our simulations with prevalence $V = 0.01$, we substitute odds ratios for relative risks. Additionally, we set the constant $A = \frac{V}{\mathbb{E}\left[\prod_{m=1}^{M} OR_m^{x_m}\right]}$ so that cases are sampled at the correct prevalence. If we assume all $M$ SNPs have a constant odds ratio $OR$ and allele frequency $p$, then $A = \frac{V}{\exp(-(Mp)^2 + ((Mp(1-p))^2(\log OR) + Mp)^2)}$. This generates the correct prevalence among randomly sampled controls, whereas simply mean-centering the genotypes or estimating the denominator as $OR^{\sum \mathbb{E}[x_m]}$ may produce inflated prevalences depending on model parameters. Additional information regarding estimation of $A$ and appropriateness of substituting odds ratios for relative risks is shown in S1 Text.

**Logistic regression.** The logistic regression model describes the log odds of disease risk as a linear function over $M$ SNPs. Rearranging this function, the disease risk can be expressed as a sigmoid function over the log odds. The effects $\beta_m$ are interpreted as $\log OR_m$ where $OR_m$ is the odds ratio of SNP $m$ with respect to disease $y$. To ensure the fraction of cases in a random sample equals the desired prevalence $V$, the intercept term $\beta_0$ is set to $-\log\left(\frac{1}{V} - 1\right)$, and genotypes in $X$ are mean-centered to 0. For a particular individual $x$,

$$P(y = 1|x) = \frac{1}{1 + \exp[-(\beta_0 + \sum_{m=1}^{M} \beta_m(x_m - \mathbb{E}[X_m]))]} \tag{2}$$

**Liability threshold model.** The liability threshold model assumes that case/control labels are assigned according to a hidden continuous liability score. Here $\beta_m$ no longer corresponds to an odds ratio, but rather an effect size on the standard normal liability scale. The fraction of variance explained by the PRS $X^T \beta$ is subtracted from the total variance 1. Individuals are assigned to cases if their liability scores pass a threshold $T$ on the standard normal distribution. If $\Phi$ denotes the standard normal cumulative distribution function, then $T$ is placed such that $1 - \Phi(T)$ equals the prevalence of the disease. Likewise, for any individual $x$, the disease risk is denoted by the probability of surpassing $T$ given that only the value of the PRS $x^T \beta$ is known.

$$P(y = 1|x) = \Phi\left(\frac{(x - \mathbb{E}[X])^T \beta - T}{\sqrt{1 - \mathrm{Var}(x^T \beta)}}\right) \tag{3}$$

We evaluated the existing BUHMBOX method in homogeneous and heterogeneous cohorts simulated from each of these GWAS models, with heterogeneous cases comprising a mixture of true cases and controls.

## CLiP: Correcting for negative correlation bias

A central assumption of the hypothesis test in Han et al. [35] is that SNPs conferring risk for a disease are uncorrelated among cases for the disease as well as controls. However, the authors prove this for only a multiplicative binary model, in which an individual's risk is the product of odds ratios of probability of disease for associated SNPs. GWAS are commonly interpreted as either a logistic or liability threshold model, both of which are generalized linear models with similar S-shaped inverse link functions over a sum of SNP effects. Particularly in the liability threshold model, the additive contribution of SNP effects to a thresholded score suggests that among individuals ascertained on that thresholded score, there may be correlatedness between those additive effects.

CLiP calculates the same heterogeneity score as previous work [35], but adjusts the null distribution to account for expected correlations between SNPs when the cohort is homogeneous and generated from a liability threshold model. The test is performed over a genotype matrix

$X$ comprising $N$ cases and $M$ SNPs counting the number of risk-alleles, as well as a matrix of controls $X^0$ with $N^0$ individuals. The SNPs included in $X$ are typically those reported in summary statistics at unique loci, and should be selected so that there is little LD between them and their correlation among controls is near 0. Pairwise SNP correlations are calculated over cases and controls separately and stored in $R$ and $R^0$ respectively. These correlations are then compared against their null expected values. If we interpret controls as population controls in that they are sampled from the full liability distribution, then the expected correlation among controls $\mathbb{E}[R^0_{jk}]$ is always 0 as SNPs are sampled independently. This modified heterogeneity score is computed as follows:

$$S_{CLiP}(X, X^0) = \frac{\sum_{j=1}^{M} \sum_{k=j+1}^{M} w_j w_k (R_{jk} - R^0_{jk} - \mathbb{E}[R_{jk} - R^0_{jk}])}{\sqrt{\frac{N + N^0}{NN^0}} \sqrt{\sum_{j=1}^{M} \sum_{k=j+1}^{M} w_j^2 w_k^2}} \tag{4}$$

where if $p_j$ and $\gamma_j$ are the allele frequency and odds ratio, respectively, for SNP $j$,

$$w_j = \frac{\sqrt{p_j(1 - p_j)}(\gamma_j - 1)}{(\gamma_j - 1)p_j + 1} \tag{5}$$

The score $S_{CLiP}$ is a weighted sum of differences in correlation between cases and controls, to account for prior sources of SNP-SNP correlation such as ancestry. A high score resulting from a bias towards positive correlations would suggest the presence of subtypes with differing ascertainment for the included risk-alleles, and thus heterogeneity. The weights are intended to adjust the score's sensitivity to certain SNPs based on their allele frequency $p$ and odds ratio $\gamma$, with larger odds ratios and frequencies close to 0.5 producing greater weights. The BUHM-BOX score is shown by Han et al. [35] to be asymptotically standard normally distributed under the null as sample sizes increase. CLiP modifies the score by shifting the expected null score from 0 to a negative value expected of homogeneous cases under the liability threshold model. This amounts to subtracting a constant from the score, which does not change the variance, ensuring $S_{CLiP}$ remains a valid Z-score test.

The expected value of the correlation $\mathbb{E}[R_{jk}]$ between two SNPs $X_j$ and $X_k$ in homogeneous cases can be calculated from the individual expectations comprising the correlation. We assume the expected value among controls, $\mathbb{E}[R^0_{jk}]$, is zero, provided that the SNPs comprising the PRS are not in LD.

$$R_{jk} = \frac{\mathbb{E}[X_j X_k] - \mathbb{E}[X_j]\mathbb{E}[X_k]}{\sqrt{\mathbb{E}[X_j^2] - \mathbb{E}[X_j]^2} \sqrt{\mathbb{E}[X_k^2] - \mathbb{E}[X_k]^2}} \tag{6}$$

We use Bayes theorem to calculate each of these expectations given the individuals are cases $(Y = 1)$. This can be done efficiently over SNPs, which take on discrete values.

$$\mathbb{E}[X_j | Y = 1] = \sum_{j \in \{0,1,2\}} X_j p(X_j | Y = 1) = \frac{\sum_{j \in \{0,1,2\}} X_j P(Y = 1 | X_j) P(X_j)}{\sum_{j \in \{0,1,2\}} P(Y = 1 | X_j) P(X_j)}$$

$$\mathbb{E}[X_j^2 | Y = 1] = \sum_{j \in \{0,1,2\}} X_j^2 p(X_j | Y = 1) = \frac{\sum_{j \in \{0,1,2\}} X_j^2 P(Y = 1 | X_j) P(X_j)}{\sum_{j \in \{0,1,2\}} P(Y = 1 | X_j) P(X_j)} \tag{7}$$

$$\mathbb{E}[X_j X_k | Y = 1] = \sum_{j \in \{0,1,2\}} \sum_{k \in \{0,1,2\}} X_j X_k p(X_j, X_k | Y = 1) = \frac{\sum_{j \in \{0,1,2\}} \sum_{k \in \{0,1,2\}} X_j X_k P(Y = 1 | X_j, X_k) P(X_j, X_k)}{\sum_{j \in \{0,1,2\}} \sum_{k \in \{0,1,2\}} P(Y = 1 | X_j, X_k) P(X_j, X_k)}$$

The case probabilities conditioned on SNP values $P(Y = 1|X_j)$ are calculated from the liability threshold model in Eq 3. As SNP associations are typically reported as odds ratios corresponding to a logistic regression model, we convert these to effects on the liability scale using Eq 28 in the Methods section, based on Gillett et al. [8].

## CLiP-X: Heterogeneity detection with quantitative predictors

While heterogeneous subtypes may occur in transcriptome-wide association studies, the heterogeneity score cannot be computed directly over continuously distributed gene expression variables rather than discrete SNPs. In CLiP, the weights $w$ are important for scaling the contributions of individual SNPs to the final heterogeneity Z-score, and they are dependent on risk-allele frequencies and odds ratios, quantities not strictly defined for continuous variables. In the case of binary variables, higher weights are assigned to SNPs with more extreme risk-allele frequencies as well as effect sizes, as these variables are more likely to generate highly positive correlations in the presence of heterogeneity. Here we generalize this weighting scheme to accommodate arbitrarily distributed continuous input variables, which may be applied in particular to expression analyses.

## CLiP-X simulation procedure

To fully simulate expression variables as modeled in transcriptome-wide association, expression predictors are generated from a linear model of randomly sampled genotypes, rather than directly sampling expression. Although the input into CLiP-X includes only the expression variables, explicitly modeling the genotype layer allows for inclusion of prior correlations resulting from SNPs associated with multiple transcripts, rather than from ascertainment by the liability threshold model.

For a single case/control phenotype, transcript effect sizes $\alpha$ are fixed to a single value so that the variance explained of all modeled transcripts is a desired value. Likewise, genotype-transcript effect sizes $\beta$ are also fixed so that variance explained of each transcript by genomic variants is a second specified value. Although fixing effect sizes at the genotype-transcript layer is admittedly unrealistic, the results are only simplified when these interactions are removed, with no interactions reducing to expression sampled from the standard normal distribution. Cases are determined according to the liability threshold model. For individuals in transcript matrix $Z$, a hidden quantitative liability score $Y^*$ is calculated, with the variance of error $\epsilon$ set so that $Y^*$ has a total variance of 1. The observed case/control label $Y$ is set according to whether $Y^*$ passes the liability scale threshold $T$, which is placed on the standard normal distribution so that affected individuals constitute a prevalence of 0.01.

$$
\begin{aligned}
Y^* &= Z^T \alpha + \epsilon \\
Y &= \begin{cases} 1 & \text{if } Y^* \geq T \\ 0 & \text{if } Y^* < T \end{cases}
\end{aligned}
\tag{8}
$$

To generate cases and controls, we iteratively generate batches of transcripts by random sampling, and compile those that pass or fail the threshold cutoff into case and control cohorts. We generate heterogeneous cohorts by concatenating simulated cases and controls, with the fraction of cases set to 0.5 for simplicity. A full description of the simulation procedure is provided in S2 Text and illustrated in S14 Fig. Note that the variance of the random noise $\epsilon$ in Eq 8 is determined by the desired total variance explained by the simulated expression variables

$V_E^2$:

$$Var(\epsilon) = \frac{1 - V_E^2}{V_E^2} Var(Z^T \alpha) \tag{9}$$

**Characterizing correlations between continuous variables.** Given $N \times L$ matrices of quantitative expression measurements $Z$ among cases and $Z^0$ among controls, we would like to determine whether $Z$ comprises a homogeneous or heterogeneous set of cases as generated in S2 Text. When $Z$ is heterogeneous, we assume the individuals in $Z$ can be assigned to one of two subtypes: one sampled according to the liability threshold model for the simulated phenotype, and one sampled randomly as controls. For a given predictor indexed by $j \in [1, \ldots, L]$, assume $Z_{ij}$ is sampled according to a mean and variance specific to the subtype of individual $i$, denoted by $Z_i^+$ for the case subtype and $Z_i^-$ for the control subtype. The distribution of the variables need not be discrete or even normally distributed, as the heterogeneity score is computed from correlations, which in turn rely only on the mean and variance of the input variables. Therefore the score can be calculated assuming any probability distribution provided that the mean and standard deviation are obtainable. For an arbitrary probability distribution $\mathcal{D}$ parameterized by its mean and standard deviation, we have:

$$\begin{aligned} X_j^+ &\sim \mathcal{D}(\mu_j^+, \sigma_j^+) \\ X_j^- &\sim \mathcal{D}(\mu_j^-, \sigma_j^-) \end{aligned} \tag{10}$$

Assume that the proportion of individuals belonging to the group $-$ is $\pi$. For a homogeneous group of cases, $\pi = 0$, and our simulations assume $\pi = 0.5$ for heterogeneous cases, but in practice this proportion is unknown. Incorporating this proportion allows the redefinition of expectations over the entire cohort as weighted sums of the expectations over the subgroups. The expected correlation evaluated over the entire group can then be calculated according to within-group expectations:

$$\begin{aligned} r_{jk} &= \frac{\mathbb{E}[Z_j Z_k] - \mathbb{E}[Z_j]\mathbb{E}[Z_k]}{\sqrt{Var(Z_j)}\sqrt{Var(Z_k)}} \\ &= \frac{\mathbb{A}_\pi(\mathbb{E}[Z_j^- Z_k^-], \mathbb{E}[Z_j^+ Z_k^+]) - \mathbb{A}_\pi(\mu_j^-, \mu_j^+)\mathbb{A}_\pi(\mu_k^-, \mu_k^+)}{\sqrt{\mathbb{A}_\pi(\mathbb{E}[(Z_j^-)^2], \mathbb{E}[(Z_j^+)^2]) - \mathbb{A}_\pi(\mu_j^-, \mu_j^+)^2} \cdot \sqrt{\mathbb{A}_\pi(\mathbb{E}[(Z_k^-)^2], \mathbb{E}[(Z_k^+)^2]) - \mathbb{A}_\pi(\mu_k^-, \mu_k^+)^2}} \end{aligned} \tag{11}$$

where $\mathbb{A}_\pi(x, y) = \pi x + (1 - \pi)y$.

**Definition of weights for continuous variables.** We would like to make use of these expectations over correlations by incorporating them as weights in the heterogeneity score. As predictors with high mean differences between subgroups and high effects are expected to contribute more signal to the score, weighting them higher than other predictors will increase power to detect heterogeneity. Therefore, we would like to define a set of weights $w_{jk}$ for each expected $r_{jk}$.

We derive the weights for continuous variables in an analogous manner to Han et al. [35], by taking the derivative of the expected sample correlation with respect to $\pi$ at the null value, $\pi = 0$.

$$w_{jk} = \left. \frac{\partial}{\partial \pi} r_{jk} \right|_{\pi=0} \tag{12}$$

To facilitate calculation of $\mathbb{E}[Z_j^+ Z_k^+]$ and $\mathbb{E}[Z_j^- Z_k^-]$ in Eq 11, we assume as in [35] that within a subgroup of cases or controls, the correlation between any two predictors, even those associated with the phenotype, is approximately zero. This allows us to express expectations of products as products of expectations. Note that this does not mean that correlations over the entire cohort $\mathbb{E}[Z_j Z_k]$ are zero: these correlations are calculated inclusive of all subgroups, and their nonzero correlations are what determines the heterogeneity score. While we demonstrate in the Results that theoretically and by simulation this assumption is violated in logistic and liability threshold models, we found it to be nevertheless a convenient method to estimate the weights $w_{jk}$.

Given the assumption of no correlation within subgroups, the correlation between two variables $Z_{\cdot j}$ and $Z_{\cdot k}$ can be expressed as the following. For further details on the derivation, please see S1 Text.

$$w_{jk} \quad = \frac{\mu_j^+ \mu_k^+ - \mu_j^+ \mu_k^- - \mu_j^- \mu_k^+ + \mu_j^- \mu_k^-}{\sigma_j^- \sigma_k^-} \tag{13}$$

$$= \frac{(\mu_j^+ - \mu_j^-)(\mu_k^+ - \mu_k^-)}{\sigma_j^- \sigma_k^-} \tag{14}$$

The same weights defined in Han et al. [35] for Bernoulli variables is a special case of this general formulation. These weights can now be substituted into the heterogeneity score.

Additionally, in practice we do not know the value of $\mu_j^+$ because the membership of individuals in each of the subsets is unknown. However, we do know the mean values of the heterogeneous case group as a whole which we denote as $\mu_j$. We can use this value as an approximation for $\mu_j^+$, and calculate an approximate weight:

$$\hat{w}_{jk} = \frac{(\mu_j - \mu_j^-)(\mu_k - \mu_k^-)}{\sigma_j^- \sigma_k^-} \tag{15}$$

We can also quantify the errors we are making by this approximation. We have the following relationship for any distribution of the genotype random variables:

$$\mu_j = \mathbb{A}_\pi(\mu_j^-, \mu_j^+) \tag{16}$$

The approximation in Eq 15 will attenuate the magnitude of $\mu_j^+$ with respect to the true value of the weight. However, we also see that:

$$\frac{\hat{w}_{ij}}{w_{ij}} = \frac{[\mathbb{A}_\pi(\mu_j^-, \mu_j^+) - \mu_j^-][\mathbb{A}_\pi(\mu_k^-, \mu_k^+) - \mu_k^-]}{(\mu_j^+ - \mu_j^-)(\mu_k^+ - \mu_k^-)} = \pi^2 \tag{17}$$

As each weight is scaled by a constant factor, their relative magnitudes are unchanged. Consequently, the heterogeneity score for continuous input variables does not change after this approximation. Thus we can still achieve optimal estimates of heterogeneity despite lacking access to the true mean for the underlying case subgroup.

## CLiP-Y: Heterogeneity detection in quantitative phenotypes

The basic CLiP test for heterogeneity relies on differential enrichment of SNP effect sizes or odds ratios across subtypes, and thus requires ascertainment for cases. But one can presume that heterogeneity exists in quantitative phenotypes as well; e.g., are there distinct genetic mechanisms predisposing individuals to being tall? But extending this method to quantitative

phenotypes presents a challenge as there is no dichotomous delineation between cases and controls. A naive solution may be to pick an arbitrary z-score as a threshold and denote samples who score higher as "cases" and those lower as "controls." This introduces a trade-off between sample size and signal specificity, as lowering this threshold provides more samples for the correlation analysis but also introduces more control-like samples which will attenuate SNP associations, and the correlations themselves. A more principled method would allow for the inclusion of all continuous samples, but give higher weight to those with large polygenic SNP scores. Thus we propose to score heterogeneity by a weighted correlation with phenotypes serving as a measure of the importance of a sample in the case set. These weights determine the degree to which individuals count as a "case", and therefore their contribution to the total heterogeneity score of the genotype matrix. Artificially creating the two groups by applying a hard threshold over the quantitative phenotype values is a special case of this method with a step function as the weighting scheme, equally weighting all individuals above the threshold "step."

## CLiP-Y simulation procedure

Here SNPs as input predictors are sampled directly from binomial distributions with fixed minor allele frequency of 0.5. The quantitative phenotype $Y$ is calculated from the PRS score with normally distributed noise added according to the desired PRS variance explained. As in the CLiP-X simulation procedure, we generate heterogeneous cohorts by concatenating a subset of cases and controls together into a single putative set of cases. For quantitative phenotypes, the "control" subset is generated so that the quantitative phenotype value is simply sampled from the normal distribution with zero PRS variance explained. A more detailed description of the simulation procedure is provided in S3 Text.

**Definition of individual weights by phenotype values.** We define a weight over individuals based on phenotype such that those with higher weight contribute more to the heterogeneity score. For a cohort of $N$ individuals let $X_{ij} \in \{0, 1, 2\}$ be the number of risk alleles of SNP $j$ in individual $i$, and let $Y = (y_1, \ldots, y_N)$ be values of the quantitative trait. We introduce a normalized weight vector across the $N$ individuals defined as $\phi \in \mathbb{R}^N$ such that $\forall i$, $\phi_i \geq 0$ and $\sum_i^N \phi_i = 1$. For an arbitrary function $\mathcal{F}$, we define $\phi \equiv \phi(\mathcal{F})$, where the weight values would reflect normalized scaling of the trait $\phi_i = \frac{\mathcal{F}(y_i)}{\sum_j \mathcal{F}(y_j)}$. Dichotomous, case/control weighting is the special case of:

$$\mathcal{F}^{01}(y_i) = \begin{cases} 1 & case \\ 0 & control \end{cases}$$

Uniform weighting is obtained by $\mathcal{F}^1(y_i) \equiv 1$. To obtain the optimal weight function which most clearly contrasts the scores of heterogeneous and homogeneous cohorts, we tested several possible functions and also performed a local search over polynomials of degree 2, 4, and 6 by iteratively updating and testing the performance of individual polynomial coefficients. This local search is described in detail in S4 Text. First, a small number of homogeneous and heterogeneous cohorts are generated as described before. These serve as the training data by which the weight function is optimized. All weight functions are applied over the raw phenotype values directly, or their conversion to percentiles in the sample distribution, in the range [0, 1]. After initially randomizing a set of coefficients, at each iteration, a coefficient is randomly selected and incremented by a random quantity sampled from a normal distribution. The resulting polynomial is tested against the training data, and the change to the coefficient is

kept if the difference in score between heterogeneous and homogeneous cohorts increases. After a set of high-performing weight functions are selected, they are each evaluated against a larger sample of validation data comprising homogeneous and heterogeneous cohorts as before. Of these candidates, the polynomial that performs best on the validation data is selected.

**Definition of weighted correlations.** To compute correlations we define, for each SNP $j$, a random variable $u_j^\phi$ with values in $\{0, 1, 2\}$ by sampling from the genotypes of the sample cohort $X_j$ with probability equal to the weight $\phi_i$ assigned to each individual $i$. Rather than calculate the correlations directly over SNPs in $X$, we now calculate correlations over these random variables. We omit the superscript $\phi$ in $u^\phi$ when it is clear from context. For a single SNP $j$, we define the weighted mean value across $N$ individuals as:

$$\mathbb{E}[u_j] = \sum_{i=1}^{N} \phi_i X_{ij} \tag{18}$$

Between two SNPs $j$ and $k$, we define the weighted covariance as:

$$\begin{aligned} \mathrm{Cov}(u_j, u_k) \quad &= \mathbb{E}[(u_j - \mathbb{E}[u_j])(u_k - \mathbb{E}[u_k])] \\ &= \sum_{i=1}^{N} \phi_i (x_{ij} - \mathbb{E}[u_j])(x_{ik} - \mathbb{E}[u_k]) \end{aligned} \tag{19}$$

We define the weighted correlation matrix $R^\phi$ for any weighting $\phi$ as:

$$\begin{aligned} R_{jk}^\phi \quad &= \mathrm{Corr}(u_j^\phi, u_k^\phi) \\ &= \frac{\mathrm{Cov}(u_j^\phi, u_k^\phi)}{\sqrt{\mathrm{Cov}(u_j^\phi, u_j^\phi)\mathrm{Cov}(u_k^\phi, u_k^\phi)}} \end{aligned} \tag{20}$$

The heterogeneity score tallies the entries of the upper-triangular correlation matrix for the phenotype-weighted individuals $R^{\phi(\mathcal{F})}$. As we now lack a held-out set of controls to cancel the contribution of correlations unrelated to the phenotype, we instead calculate a conventional correlation uniformly weighted across all individuals $R_0 \equiv R^{\phi(\mathcal{F}^1)}$. Additionally, we introduce a scaling factor of $\sqrt{(\sum_{i=1}^{N} \phi_i^2) - \frac{1}{N}}$ to correct for the change in variance resulting from re-weighting the correlation according to individual weights $\phi_i$. These changes produce the following preliminary heterogeneity score for quantitative phenotypes:

$$Q = \frac{\sum_{j=1}^{M} \sum_{k=j+1}^{M} R_{jk}^\phi - R_{jk}^0}{\sqrt{(\sum_{i=1}^{N} \phi_i^2) - \frac{1}{N}}} \tag{21}$$

Lastly, we incorporate into the test statistic $Q$ a weighting scheme over SNPs as described in Han et al. [35]. This second set of weights $w \in \mathbb{R}^M$ is introduced to correct for larger contributions to the score by SNPs with large effect sizes or risk allele frequencies close to 0.5. These weights apply to SNPs, and should not be confused with the weights $\phi$ over individuals. For each SNP $j$, we define $p_j^\phi \equiv \frac{\mathbb{E}[u_j^\phi]}{2}$, the sample allele frequency weighted by the individual phenotype, as opposed to the unweighted allele frequency $p_j^0 \equiv p_j^{\phi(\mathcal{F}^1)}$. The contribution of SNP $j$ to

the heterogeneity score is then scaled by

$$w_j^\phi = \frac{\sqrt{p_j^0(1-p_j^0)}(\gamma_j^\phi - 1)}{((\gamma_j^\phi - 1)p_j^0 + 1)} \tag{22}$$

where

$$\gamma_j^\phi = \frac{p_j^\phi(1-p_j^\phi)}{p_j^0(1-p_j^0)} \tag{23}$$

is a weighted generalization of an odds ratio. These weights are analogous to those found in Han et al. [35], where given case allele frequency $p_j^+$, control allele frequency $p_j^0$, and sample odds ratio $\gamma_j = \frac{p_j^+(1-p_j^+)}{p_j^0(1-p_j^0)}$, the weight is

$$w_j = \frac{\sqrt{p_j^0(1-p_j^0)}(\gamma_j - 1)}{((\gamma_j - 1)p_j^0 + 1)} \tag{24}$$

Combining these intermediate calculations, the heterogeneity test statistic for continuous phenotypes is:

$$S_{CLIP-Y}(X, y) = \frac{\sum_{j=1}^{M} \sum_{k=j+1}^{M} w_j^\phi w_k^\phi (R_{jk}^\phi - R_{jk}^0)}{\sqrt{\sum_{i=1}^{N} \phi_i^2 - \frac{1}{N}} \sqrt{\sum_{j=1}^{M} \sum_{k=j+1}^{M} (w_j^\phi)^2 (w_k^\phi)^2}} \tag{25}$$

For high $N$, this test statistic approaches the standard normal distribution, and can be evaluated as a z-score hypothesis test.

Note that even when applying a dichotomous weighting scheme, dividing the cohort with quantitative phenotypes into artificial cases and controls, CLiP-Y still differs slightly from a direct application of the case/control score. If a dichotomous weight function produces $N^\phi$ artificial cases, the scaling factor $\frac{1}{\sqrt{\sum_{i=1}^{N} \phi_i^2 - \frac{1}{N}}}$ simplifies to $\sqrt{\frac{N^\phi N}{N - N^\phi}}$ instead of the slightly smaller $\sqrt{\frac{N^\phi N}{N + N^\phi}}$ in the original case/control score. This corrects for the slight reduction in variance of $R_{jk}^\phi - R_{jk}^0$ because these differently-weighted correlations are taken over a single cohort of individuals rather than disjoint sets of cases and controls. In practice, we find this correction factor performs very well in scaling the test statistic variance to 1.

## Evaluating heterogeneity in SCZ

We applied CLiP to test for heterogeneity in case/control data for schizophrenia collected by the Psychiatric Genomics Consortium (PGC). The data comprise in total roughly 23,000 cases and 28,000 controls and was the subject of a 2014 meta-analysis reporting 108 schizophrenia-associated loci [36]. These cohorts were collected largely from European populations in the United Kingdom, Sweden, Finland, United States, Australia, and others, along with populations in Portugal, Bulgaria, and Israel. Most cohorts were diagnosed with schizophrenia clinically according to standards in DSM-IV. The average sex distribution of all cohorts is 60.4% male, with a minimum of 39.4% (dubl) and maximum of 91.8% (lacw). Sex information for the cohort s234 is missing. Further information for each cohort can be found in Ripke et al. [36]. We would like to test whether heterogeneity suggested from clinical observation is also detectable at the level of the PRS comprising these loci. The PGC data is an aggregate of

cohorts collected from many studies conducted in different populations. Therefore a test for heterogeneity over the all cohorts is likely to be confounded by ancestry stratification or batch effects between cohorts. We attempt to circumvent these confounding variables by applying GWAS meta-analysis methods to CLiP scores evaluated over individual cohorts, as well as evaluating the p-value of the sum of all CLiP scores. As the CLiP score of each cohort has a variance of 1 under the null, the distribution of their sum has a standard deviation of $\sqrt{N}$ if $N$ is the number of cohorts in the sum. The expectation of the sum is the sum of the calculated expected values of each cohort. To evaluate the significance of CLiP Z-scores across individual cohorts, we applied Fisher's method for summing p-values [47].

$$\chi^2 = -2\sum_{i=1}^{K}\log p_i \tag{26}$$

where $K$ is the total number of cohorts and $p_i$ is the p-value of the CLiP heterogeneity score for cohort $i$. The p-value of this test statistic is evaluated on a chi-square distribution with $2K$ degrees of freedom. Additionally, we calculated the meta-analysis Z-score of the CLiP score in a manner analogous to the conventional GWAS approach, but with a 1-tail test for highly positive scores only. The meta-analysis Z score is calculated according to

$$Z_i = \text{sign}(Z_{CLiP})\Phi^{-1}(1 - p_i)$$

$$Z = \frac{\sum_{i=1}^{K} Z_i n_i}{\sqrt{\sum_{i=1}^{K} n_i^2}} \tag{27}$$

where $Z_{CLiP}$ is the CLiP Z-score evaluated against the expected score with a standard deviation of 1, and $n_i$ is the sample size of cohort $i$.

While hidden batch effects may still be present in single cohorts, these effects are not expected to bias heterogeneity scores to the same extent as scoring cases and controls combined across all cohorts. By calculating the difference in summed correlations between cases and controls of a single cohort, CLiP cancels the effect of confounding heterogeneity present in both cases and controls provided that this heterogeneity is near-equally represented in both sets. Combining all cohorts from different populations introduces more confounding heterogeneity into the analysis by virtue of the increased diversity of the combined populations. Additionally, the differing sizes of the cohorts ensures that the representation of populations differs between the cases and controls.

## Application to genomic data sets

We applied CLiP to GWAS data from the PGC, phased and imputed using SHAPEIT [48] and IMPUTE2 [49], a pipeline with similar or better accuracy compared to other tools according to a recent evaluation [50]. Imputation was performed using the 1000Genomes Phase 3 reference panel. Roughly half of the PGC cohorts were mapped with assembly NCBI36, and the SNP coordinates of these data sets were converted to GRCh37 using the LiftOver tool in the UCSC genome browser database [51]. Individuals were excluded from further analysis if their percentage of missing data was greater than 0.1 in the 1 Mb region flanking each SNP. Additionally, of the 108 associated SNPs and indels reported in Ripke et al. [36], three SNPs located on the X chromosome were excluded, three were excluded because they are not listed in the 1000Genomes Phase 3 reference panel, one was excluded due to low variance in many individual study cohorts, and one was excluded due to mismatching alleles between reported

summary statistics and the reference panel, for a total of 100 variants included in the heterogeneity analysis.

To accurately estimate expected heterogeneity scores, the odds ratios reported in Ripke et al. [36] must be converted to effect sizes on the liability scale. We apply an approximate method reported by Gillett et al. [8] to convert for variant $j$ an odds ratio $OR_j$ to the liability effect $\beta_j$:

$$\beta_j \simeq \Phi^{-1}(F_{Logistic}(\log\frac{V}{1-V} + \log OR_j)) - \Phi^{-1}(V) \tag{28}$$

where V is the disease prevalence (0.01 for schizophrenia), and $F_{Logistic}(x) = \frac{1}{1+\exp(-x)}$.

We likewise ran SHAPEIT and IMPUTE2 with the 1000Genomes Phase 3 reference panel to perform imputation of UK Biobank data. We obtained summary statistics for 119 associated SNPs for the neuroticism sum-score from Nagel et al. [44], using a p-value cutoff of $1 \times 10^{-8}$. Of these, we were able to match 108 SNPs to the 1000 Genomes Phase 3 reference panel. We estimated the variance explained of these SNPs with the sum-score to be 0.1 as reported in Supplementary Figure 19 in Nagel et al. [44]. Lastly, to speed up computation time, we estimated the expected CLiP-Y score for homogeneous cohorts by simulating 400 cohorts of equivalent size to the neuroticism cohort in lieu of calculating the expected correlation for all pairs of SNPs, which requires numeric integration. In panel A of S9 Fig and panel A of S10 Fig, mean scores obtained by sampling fit closely the calculated expected scores across all weight functions over only 20 simulated trials for each set of parameters.

## Supporting information

**S1 Text. Supplementary methods and analyses.**
(PDF)

**S2 Text. Sampling procedure for heterogeneous cases with quantitative predictors.** We assume the input predictors are normally distributed gene expression transcripts generated from linear functions of sampled binomial SNPs and total SNP variance explained $V_G^2$. Then cases and controls are sampled based on a liability threshold model on these gene expression variables.
(PDF)

**S3 Text. Sampling procedure for heterogeneous PRS cohorts with quantitative phenotypes.** In lieu of explicit case/control labels by which to generate heterogeneous cohorts, we define 'cases' as individuals whose phenotypes are a function of their PRSs, whereas 'controls' have phenotypes that are sampled completely randomly from the same distribution.
(PDF)

**S4 Text. Local search for optimal polynomial weight function $\phi$ for use with CLiP-Y.** This weight function is applied over individuals according to their phenotypes as a percentile of the phenotype distribution in the domain [0, 1], and determines the contribution of individuals to the weighted correlation substitution in CLiP-Y. Optimization is performed over a small set of discovery simulated cohorts, and validated in a larger set of simulated cohorts. A set of candidate functions which pass a fixed threshold are stored, and of these the function which performs best on the larger validation set is selected.
(PDF)

**S1 Fig. Sample disease prevalences of cohorts simulated according to logistic and liability threshold models.** Each sample comprises 100,000 individuals with randomly sampled

genotypes, and case/control status was assigned according to either a logistic or liability threshold model with a desired prevalence of 0.01. The resulting fraction of individuals assigned to cases by each model is plotted. Colors indicate a fixed odds ratio assigned to 10 SNPs, with each value simulated over 10 trials.
(PDF)

**S2 Fig. CLiP performance on simulated control, homogeneous, and heterogeneous cohorts. Shown are heterogeneity scores resulting from different cohort sizes and genetic architectures. (A)** Case/control cohorts generated from logistic or Risch models with 100 diploid SNPs with allele frequency of 0.2 and odds ratio of 1.2. A prior prevalence of 0.01 is assumed for both models. Scores of logistic cohorts exhibit a negative bias resulting from correlations between SNPs not seen in the Risch model. **(B)** Case/control cohorts generated from logistic or Risch models with 100 diploid SNPs with allele frequency of 0.2 and odds ratios of 1.06, and prior prevalence of 0.01. With smaller effect sizes, the negative bias in the logistic model disappears.
(PDF)

**S3 Fig. CLiP performance as a function of subtype fraction size.** Heterogeneity scores (y-axis) evaluated on heterogeneous cohorts comprising a mixture of true cases and controls at different proportions (x-axis). Colors indicate the total cohort size. The X-axis indicates the fraction of individuals that are true cases. When the fraction is 0, the cohort contains only controls, and all expected correlations are 0, producing a heterogeneity score of 0. When the fraction is 1, the cohort contains only cases, and produces a highly negative score due to negative correlations between all pairs of SNPs. As expected, a mixture of cases and controls produces positive scores, with the peak score occurring when the cohort is split evenly. More detailed results of this set of simulations are shown in S2 Table. All tests were conducted with a SNP variance explained of 0.05.
(PDF)

**S4 Fig. CLiP is robust to confounding heterogeneity with an $F_{st} < 0.05$.** CLiP corrects for confounding heterogeneity such as ancestry by calculating the difference in SNP-SNP correlations between cases and controls. Any confounding patterns that are present in controls as well as cases are then canceled from the score. **(A)** Simulated homogeneous and heterogeneous cohorts in which both cases and controls are sampled from two sub-populations with an $F_{st}$ specified by color. **(B)** Power (dotted) and specificity (solid) over 20 trials. At high $F_{st}$ values, both homogeneous and heterogeneous case scores are attenuated towards zero, and while power remains high, specificity begins to decline with values of $F_{st}$ greater than 0.05.
(PDF)

**S5 Fig. CLiP performance as a function of number of distinct subtypes. Heterogeneity scores (y-axis) evaluated on simulated heterogeneous cohorts with disjoint sub-phenotypes.** Performance is shown a function of the fraction of SNP effects unique to a particular sub-phenotype (x-axis). Colors indicate the number of sub-phenotypes. Simulations were performed with 50,000 simulated cases and 50,000 controls, and a total SNP variance explained over all sub-phenotype PRSs set to 0.05. More detailed results of this set of simulations are shown in S3 Table.
(PDF)

**S6 Fig. Creating CliP-Y weight functions using step functions.** To demonstrate that optimal quantitative weight functions for heterogeneity are concave functions, two interval indicator functions in [0, 1], an increasing one for [$x$, 1] (x-axis) and a decreasing one for [0, $y$] (y-axis)

are combined so that their sum is the tested weight function. Each bin on the axes represents a transition point for the two step functions. The heterogeneity score is tested against a single homogeneous cohort, so optimal scores should be those that are most negative. **(A)** The best scores are those where *x* is low on the PRS percentile scale but not 0, while *y* is high on the PRS percentile scale but not 1. This coincides with the optimal polynomial functions obtained by a local search. **(B)** A zoomed view of the top left in A, showing that the optimal scores are not obtained by step functions at the periphery of the PRS distribution.
(PDF)

**S7 Fig. CLiP-Y scores for quantitative phenotypes split into artificial cases and controls by a hard threshold, as a function of cohort size.** Means and standard deviations of scores are shown as a function of sample size: **(A)** homogeneous cohorts, **(B)** heterogeneous cohorts, and **(C)** the difference (heterogeneous minus homogeneous) scores. The color gradient indicates the location of the threshold separating cases and controls. Each condition was run for 20 trials, and all cohorts were simulated with 100 SNPs and a total variance explained of 0.1.
(PDF)

**S8 Fig. CLiP-Y scores for quantitative phenotypes split into artificial cases and controls by a hard threshold, as a function of SNP variance explained.** Means and standard deviations of scores are shown as a function of total variance explained by SNPs: **(A)** homogeneous cohorts, **(B)** heterogeneous cohorts, and **(C)** the difference between scores of heterogeneous cohorts and expected homogeneous scores in A. Colors indicate the type of weight function, with blue lines indicating learned polynomial functions. Each condition was run for 20 trials, and all cohorts were simulated with 100 SNPs and a sample size of 100,000 cases and 100,000 controls.
(PDF)

**S9 Fig. CLiP-Y scores for quantitative phenotypes with weight functions over individuals contributing to the correlation, as a function of cohort size.** Means and standard deviations of scores are shown as a function of sample size: **(A)** homogeneous cohorts, **(B)** heterogeneous cohorts, and **(C)** the difference between scores of heterogeneous cohorts and expected homogeneous scores in A. Colors indicate the type of weight function, with blue lines indicating learned polynomial functions. In panel A, black dotted lines indicate calculated expected scores for each homogeneous cohort, given effect sizes and allele frequencies. In practice, these expected scores will serve as the null scores subtracted from that of test data to produce analogous results to panel C. Each condition was run for 20 trials, and all cohorts were simulated with 100 SNPs and a total variance explained of 0.1.
(PDF)

**S10 Fig. CLiP-Y scores for quantitative phenotypes with weight functions over individuals contributing to the correlation, as a function of SNP variance explained.** Means and standard deviations of scores are shown as a function of variance explained: **(A)** homogeneous cohorts, **(B)** heterogeneous cohorts, and **(C)** the difference between scores of heterogeneous cohorts and expected homogeneous scores in A. Colors indicate the type of weight function, with blue lines indicating learned polynomial functions. In panel A, black dotted lines indicate calculated expected scores for each homogeneous cohort, given effect sizes and allele frequencies. In practice, these expected scores will serve as the null scores subtracted from that of test data to produce analogous results to panel C. Each condition was run for 20 trials, and all cohorts were simulated with 100 SNPs and a sample size of 100,000 cases and 100,000 controls.
(PDF)

**S11 Fig. Additional CLiP-X scores as a function of variance explained by quantitative predictors.** Shown are controls (black), homogeneous cases (red), and heterogeneous cases (green) across different levels of variance explained by generated predictors ($V_E^2$). The predicted score for homogeneous cases (blue), calculated from summary statistics used to generate predictors from sampled SNPs and phenotypes from predictors, is the true null hypothesis of the heterogeneity test. Means and standard errors are shown for 20 trials over 10 expression variables generated by 100 SNPs.
(PDF)

**S12 Fig. Validation of expected correlations in CLiP-X between expression variables in cases.** Shown are the average standard deviations between predicted and generated correlations as a function of the case sample size used to estimate the correlation. Predicted values account for contributions from PRS thresholding and shared SNP-expression effects. Standard deviations across all pairs of variables are averaged for each experiment. The black line denotes the function $\frac{1}{\sqrt{N-3}}$, the expected standard deviation of the Fisher transformation for sample correlations. Values on the y-axis are transformed by an inverse hyperbolic function (artanh) for comparison to the Fisher transformation.
(PDF)

**S13 Fig. Heterogeneity scores of the largest PGC cohorts subdivided by inferred ancestry group.** Original whole-cohort scores are shown in black. Ancestry inference was performed using fastStructure with number of ancestry groups $k = 2$ (red), $k = 3$ (green), and $k = 4$ (blue). While subgroups of small size are subject to larger correlation error, generally the heterogeneity scores of ancestry subgroups achieves the same magnitude as the original score, suggesting ancestry stratification is not the primary source of heterogeneity in PGC schizophrenia cohorts.
(PDF)

**S14 Fig. Sampling procedure for CLiP-X: Heterogeneity detection with quantitative input variables.** Because in most scenarios the inputs will represent gene expression, we simulated transcripts as linear functions of randomly sampled genotypes, allowing for prior correlations from single genotypes associated with multiple transcripts. To generate a desired number of cases (Red), batches of random transcripts are generated repeatedly, and those individuals whose liability scores pass the threshold $T$ are concatenated to a growing list of cases. Heterogeneous cases are created by concatenating true cases with controls.
(PDF)

**S1 Table. Power calculations accompanying CLiP and BUHMBOX simulation results in Fig 3.** For both methods, the reported power is scored on true heterogeneous cases by a Z-test with mean set to the expected score of homogeneous cases. In BUHMBOX, the expected score is 0 in keeping with a multiplicative assumption across SNPs. In CLiP, the expected score is some negative value corresponding to the negative correlations between SNPs expected of cases selected from a liability threshold model, providing a boost in power. Tables A and B correspond to panels A and B of Fig 3, respectively. "Sample Power" refers to the fraction $\frac{\text{True Pos.}}{\text{False Neg.+True Pos.}}$ of 20 trials shown in Fig 3 which pass a 95% confidence interval threshold. "Expected Power" refers to the percentile of the sample distribution of true heterogeneous cases passing the 95% confidence interval threshold.
(PDF)

**S2 Table. Simulated CLiP results over additional parameters.** These simulations differ by total variance explained ($h^2$), cohort size, and percentage of individuals in the cohort that are

true cases, with the remaining individuals being simulated controls. All trials were run with 100 SNPs with a fixed uniform effect size and an allele frequency of 0.2. Shown are mean and standard deviations of 20 trials.
(PDF)

**S3 Table. Simulated CLiP results using cases generated from multiple correlated sub-phenotypes.** Entries comprise mean and standard deviations of CLiP scores evaluated over 20 trials, with a total variance explained of 0.05 and total case set size of 50000. All simulations were performed with 100 SNPs, and in independent sub-phenotypes these 100 were subdivided equally among the number of sub-phenotypes. The percentage of SNPs shared refers to the percentage of SNPs within each sub-phenotype which has a fixed effect size across all sub-phenotypes.
(PDF)

## Acknowledgments

The authors would like to thank Dr. Stephan Ripke for discussions and assistance in imputation of data from the Psychiatric Genomics Consortium.

## Web resources

A Github page for CLiP, including code to reproduce figures is available at: https://github.com/jyuan1322/CLiP.

## Author Contributions

**Conceptualization:** Jie Yuan, Itsik Pe'er.

**Data curation:** Jie Yuan, Todd Lencz.

**Formal analysis:** Jie Yuan, Henry Xing, Alexandre Louis Lamy.

**Funding acquisition:** Itsik Pe'er.

**Investigation:** Jie Yuan, Henry Xing, Alexandre Louis Lamy, Itsik Pe'er.

**Methodology:** Jie Yuan, Henry Xing, Alexandre Louis Lamy, Itsik Pe'er.

**Project administration:** Jie Yuan, Itsik Pe'er.

**Resources:** Todd Lencz, Itsik Pe'er.

**Software:** Jie Yuan, Henry Xing, Alexandre Louis Lamy.

**Supervision:** Todd Lencz, Itsik Pe'er.

**Validation:** Jie Yuan, Henry Xing, Alexandre Louis Lamy, Itsik Pe'er.

**Visualization:** Jie Yuan, Henry Xing, Alexandre Louis Lamy.

**Writing – original draft:** Jie Yuan, Henry Xing, Alexandre Louis Lamy, Itsik Pe'er.

**Writing – review & editing:** Jie Yuan, Todd Lencz, Itsik Pe'er.

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
