## [Decision Letter · Decision Letter 0]

27 Jan 2020

Dear Dr Yuan,

Thank you very much for submitting your Research Article entitled 'Leveraging correlations between polygenic risk score predictors to detect heterogeneity in GWAS cohorts' to PLOS Genetics. Your manuscript was fully evaluated at the editorial level and by independent peer reviewers. The reviewers appreciated the attention to an important problem, but raised some substantial concerns about the current manuscript. Based on the reviews, we will not be able to accept this version of the manuscript, but we would be willing to review again a much-revised version. We cannot, of course, promise publication at that time.

If you decide to revise the manuscript for further consideration at PLOS Genetics, please aim to resubmit within the next 60 days, unless it will take extra time to address the concerns of the reviewers, in which case we would appreciate an expected resubmission date by email to plosgenetics@plos.org.

[LINK]

We are sorry that we cannot be more positive about your manuscript at this stage. Please do not hesitate to contact us if you have any concerns or questions.

Yours sincerely,

Samuli Ripatti

Associate Editor

PLOS Genetics

Scott Williams

Section Editor: Natural Variation

PLOS Genetics

While the reviewers appreciate the potential novelty in the method, they also raise serious concerns about the modeling approach, clarity in the presentation of the method, its usefulness in the presence of population stratification and genotyping batch effects, the presented simulations and the applicability to the real world genetic data that need to be thoroughly addressed before reconsidering the publication of the manuscript.

Reviewer's Responses to Questions

**Comments to the Authors:**

Reviewer #1: The authors present a method to detect heterogeneity within a single GWAS sample. The main method (CLiP) is designed for case-control GWAS with SNP data, using only those SNPs that reach genome-wide significance in the GWAS to form a PRS. The authors explore two main scenarios: (1) misclassification in which the cases are comprised of true cases and controls, and (2) the cases are made up of subtypes of the diseases and controls are true controls for both subtypes. The simulations are well thought out and demonstrate the behavior of this method in a variety of scenarios. However, the authors do not spend adequate space discussing the benefit of CLiP versus prior methods with no direct comparisons of other metrics. Additionally, they do not demonstrate the distinction between multiple sources of heterogeneity, such as batch effects, population stratification, case misclassification, or clinical subtypes. This is a noted limitation of prior methods, therefore should be addressed in the main text.

Major comments:

• The Results section is currently written similarly to a Method section, with much of it detailing the simulation framework. While this is helpful, the section would also benefit from more context given to the results, with specific numbers given within the text. It is currently up to the reader to interpret the Results from the figures, which can be confusing.

• The major criticisms of prior methods in the Introduction cite the inability to distinguish the hidden factors as due to population stratification or batch effects. However, it is not clear that this method is able to distinguish from these reasons either, as the example from PGC shows that it can detect batch effects between cohorts (pooled analysis) that is not present in the summed stratified analyses.

• As CLiP is an extension of BUHMBOX, but correcting for the negative correlation bias, it would be helpful for a direct and explicit comparison, both in the simulations and the applied PGC example. For example, a figure showing the power to detect heterogeneity with and without this correction should be included in the main results.

• It is not clear from the manuscript if it is possible to distinguish between batch effects and heterogeneity by subtype/misclassification within a single sample. In the applied example given in PGC, multiple cohorts are required to distinguish between heterogeneity overall (pooled cases/controls) and heterogeneity without batch effects from the different cohorts (summed across cohort-level heterogeneity scores). Therefore, it may be misleading to conclude that this method is superior from other previous methods due to this ability.

• The majority of simulations center around a misclassification framework, in which the “cases” are comprised of both true cases and controls. The framing of the method around subtypes, with examples given for bipolar disorder and schizophrenia, is not fully justified by the results and simulations. The manuscript may be better served by expanding upon results presented in Figure 2B to support this utility.

• The application of CLiP to a schizophrenia case-control GWAS could also be expanded. While subtypes of schizophrenia are less well defined, an exploration for possible sources of heterogeneity within these cohorts would be useful for interpretation. As subtypes of bipolar disorder are better defined, if this data is available it may be a better suited example to pick apart the potential contributions of batch effects, population stratification, and heterogeneity due to subtypes.

• A discussion of the limitations for this method should be included in the Discussion. For example, what are the assumptions made for the negative correlation bias correction that may not be appropriate for some scenarios?

• As low power is cited as a prior limitation to detect heterogeneity, an analysis evaluating power should also be included in the results section to weigh the merits of this new method.

• In general, it would be helpful to expand upon the schizophrenia analysis to fully demonstrate the underlying factors of heterogeneity. This would provide more context for the usefulness of this method and what underlying issues or future directions it can uncover. A discussion of these aspects would make this manuscript more relevant for a general genetics audience and increase the likelihood of adoption.

Minor comments:

• Page 11, lines 203-204 refer to Figure 3, but it is unclear which panels they refer to. Figure 3B and Figure 3C are not referenced directly in the text.

• Page 8, line 150 has “heterogeneity” misspelled.

• There are currently no methods detailing the studies in PGC. For example, were all participants of European ancestry and similar ages? Similar diagnostic criteria? This would help in the interpretation of results.

• The word “cohort” is used for both the “cases” (comprised of true and false cases) and the overall sample size of “cases” and “controls” This can be confusing at times and alternative language should be used to distinguish between the samples.

Reviewer #2: This study presents a method called CLiP which uses the correlations between SNPs (or predicted expression levels) in case patients to dissect heterogeneous subsets in cases. This is a generalization of Han et al method. The authors’ contributions are two: (1) they showed that the correlation=0 assumption under no heterogeneity is only valid in multiplicative model but not in liability threshold model and presented a correction for that. (2) they generalized the method to a) when other predicting variables than SNPs are used and b) when continuous phenotype is of interest.

Overall, the paper is quite well written. The problem is interesting and important, and the solution is straightforward. I find contribution (2) quite interesting. Contribution (1) is also important, but there can be a bit of controversy. First, the authors say that the correlation would be negative if “logistic or liability threshold model” is true. In liability threshold model, it is clear that the correlation becomes negative. For logistic model, I didn’t understand why the correlation has to be negative. In fact, Risch’s multiplicative model can be presented as logistic model. Multiplicative nature of ORs of SNPs can be modeled as additive in logit space. This seems to be written in the latter part of Supplementary Note section 9.2, but the long derivation was hard to follow. Probably the authors can give an intuitive explanation about that? Second, I personally agree that liability threshold model more closely models the truth, but am not sure if it’s really the truth. The truth might be in-between of the two (liability threshold model and multiplicative). I feel that the popularity of liability threshold model was driven from the heritability estimation (where liability is a must for binary traits). But that’s because liability is one reasonable explanation that can model heritability, but not because liability is strictly true. In sum, no one knows if that’s true. I am not sure if there were any studies that tried to test if liability threshold model more closely models the truth in complex diseases. It would be better if the authors can do some literature search and cite some. (If there were no such studies, I think extension of CLiP can be one method in the future.) I agree that Han et al. method can be underpowered by non-multiplicative model (e.g. liability), but worry if CLiP can increase false positive rate if the truth would be in-between.

In the method, I expected some equations to derive the negative expected correlations under the liability model, but was surprised that they were skipped directly to CLiP-X, leaving relatively short contents for the main CLiP. I feel that this part is as important as CLiP-X and Y, so hope that a summary of 9.4 can be in the main method, so that readers can easily implement the CLiP method using their own code.

Minor comments:

The title is confusing. I didn’t catch what “polygenic risk score predictor” means. Given title, I expected contents related to the (currently popular) usage of PRS for disease predictions. I understand that the authors did not want to use SNP or locus to emphasize that expressions can be used, but not sure if the current title is the best choice.

The authors need to add more detailed and elaborated explanations about the “transcriptome-wide association study”. First, many geneticists might not be familiar with TWAS. The authors need to explain that there is an effort to associate the gene expression level of each gene with the phenotype. Second, to some people (like me), TWAS has double meaning: method name and study design name, so can be easily confusing.

The method depends on the independency of SNPs. Did the authors do some LD pruning? Subtracting the control correlations might help controlling for long-range LD; was this the case in real data? Or controls are actually not needed?

P9. Line 173: PRSs are expected to be zero than positive → you mean, negative?

P11. Line 209: when the contents refer to Figure 3, I didn’t find it relevant. Is this a correct figure number?

Reviewer #3: The manuscript considers a statistical method to detect phenotypic heterogeneity in GWAS cohorts by using individual-level genotype data on GWAS samples. The idea is that the correlation between genotypes at different SNPs in cases increases if the cases contain heterogeneous subgroups. The method is applied to schizophrenia data. The manuscript also extends the method to continuous traits but no real data examples are given.

I find it difficult to evaluate the novel contribution of this work because the method is not presented very clearly (see below for details) and it is unclear how big a difference to the existing method is in realistic settings. The continuous versions CLiP-X and CLiP-Y seem experimental and no real data example is given. The current real data example with schizophrenia GWAS does not convincingly show how the method could conclude whether there is or is not heterogeneity in cohorts.

About the term “controls”

Unclear whether “controls” in this paper refers to a sample from the population or to proper disease-free controls. If former, then should talk about “population controls” and make clear that these are not proper disease-free controls. Theoretically, this has an important effect on whether risk SNPs are expected to be independent in “controls”. Typically, SNPs at different loci are independent in general POPULATION that is a mixture of cases and disease-free controls. And when risk SNPs are independent in population they are not independent either in cases nor in disease-free controls.

Models:

Three disease models are mentioned: “multiplicative Risch/binary model”, “logistic model” and “liability threshold model”.

The manuscript does not define clearly “multiplicative (Risch/binary) model”, or rather what is its difference from the logistic model. This should be done in Methods.

Best proxy is line 131 saying: “a multiplicative binary model, in which an individual's risk is the product of odds ratios of probability of disease for associated SNPs”

First, risk is a product of relative risks, not odds-ratios. I assume that SNP effects are defined in terms of ORs and not RRs and so the multiplicative model should be formulated for odds of disease?

Second, there needs to be some baseline risk for an individual without any risk allele on which each risk allele adds to. That together with relative risks or odds-ratios and genotype distribution in population define the prevalence in the population. Unclear what is the population prevalence in the example of Figure 1A and whether prevalence of disease is matched between the “multiplicative model” and “logistic model”. Both prevalence and ORs used in simulations should be realistic.

Third, the logistic regression says that odds of disease increase multiplicatively with each copy of the risk allele, and multiplication happens independently across the SNPs. Thus I see no difference between “multiplicative model” and “logistic regression model” as statistical models. I assume that the difference is that the assumption about the population genotype distribution are different in settings where authors use these two models. In settings with “multiplicative model” the authors assume that risk SNPs are independent in CASES (theoretically an unrealistic assumption), whereas in settings with “logistic model” the assumption is that SNPs are independent in the whole POPULATION (realistic assumption). However, when prevalence is low (a few percent or less) then independence between SNPs in population together with the logistic regression model lead to approximate independence of SNPs in CASES (as well as in disease-free CONTROLS), see e.g. Pirinen et al. Nat Gen 44, 848–851(2012). Thus, if prevalence is realistic for typical common diseases, I would not expect a large difference from independence in cases, and hence no large negative correlations between SNPs in cases. The paper should make clear whether the theoretically correct fact of negative risk SNP correlations in cases truly has a practical effect over earlier method (BUHMBOX) when prevalence is realistic. The simulations in Figure 1A should be done so that both settings (“multiplicative” and “logistic”) have the same population prevalence and ORs, and then it should be reported whether there is a practical difference after the correction for negative correlations in cases.

Fig.2 “Colors indicate the total cohort size. All tests were conducted over 50,000 cases and 50,000 controls”

Inconsistent that sample varies and is fixed to 50k.

Methods section mentions one score: S_het, “modified heterogeneity score”. Is this what is used in all Figures? Then should use symbol “S_het” in main text to make this clear.

L.136 “conventional, case-control score for heterogeneity”

This is not defined anywhere in Methods.

L. 138 “We simulated individual data using 100 independent SNPs”

Independent in population, or in cases or in controls?

L.140 “we assumed a variance explained of 1”

Needs explanation. I assume that this means that the phenotype is deterministically defined from genotypes. ORs are now the same but do both models also lead to the same prevalence in the population?

L. 141 “We evaluated the dichotomous-trait score relative to sample sizes for both homogeneous and heterogeneous groups.”

Unclear what this means?

“case cohorts were generated by repeatedly sampling control genotypes and selecting individuals whose PRS pass a threshold corresponding to a prevalence of 0.01”

I assume that genotypes are sampled from POPULATION frequencies, not from CONTROL frequencies? Or if controls are population samples i.e. a mixture of cases and proper controls, then that should be said very clearly.

L.153 “PRS over 108 genomewide-signicant SNPs with a total variance explained of 0.034”

On which scale is this variance? And is it in population or in some case-control sample?

L.185 CliP-Y section starts: “In practice, we found converting PRSs to percentiles improved performance for all learned weight functions, possibly because percentiles limit the domain of the PRS function over which the function must be 0, and they reduce the contribution to the score calculation by extreme PRS values.”

Some kind of intro to what this section aims to do and what weight functions etc are should be stated before the sentence above, that currently is incomprehensible when first read by the reader who has not yet read the Methods that come only later.

L 293. “the presence of distinct mixtures of cases within a cohort which have been identified as cases through different PRSs.”

Unclear whether this means that the cases are identified through different PRSs or whether these cases happen to have heterogeneity and hence different distributions of PRSs?

“misclassifcation, whereby a subset of individuals are not really cases, but have been rather labeled as such despite genetically being controls”

Why there is word “genetically” there? Missclassification happens purely by mislabeling the phenotype, and is not informed by genotypes.

L 296. “but it may also suggest distinct disease etiologies, some of which are not ascertained for the PRS of interest. Second, a mixture of unobserved sub-phenotypes with distinct PRSs.”

What is the difference here? Both refer to a setting where sub-phenotypes of cases have different effect sizes (“distinct PRSs”).

L 326. “The expected correlation among controls, E [R0jk], is always 0 in practice as SNPs are sampled independently, but is included below for clarity.”

If you assume that SNPs are independent in POPULATION, and that the logistic model holds, then risk SNPs are independent neither in disease-free CONTROLS nor in CASES. (But if prevalence is low, they are approximately independent in both disease-free CONTROLS and in CASES. And if you mean by controls the population sample then the sentence is true, but uses unclear terms.)

Supplementary material:

L.759-760 For which genotype group Is prevalence B given? Odds-ratios do not directly correspond to risk, but to odds.

(27) and elsewhere would be clearer is conditioning on being case was explicitly written out.

(28) This equation is based on the ASSUMPTION that p(x_i,x_j) = p(x_i)*p(x_j) in cases. If this is the assumption of "Risch model", then that should be simply stated, and there is no need to show mathematically that the SNPs are independent if that is already assumed.

(33) Assumes Hardy-Weinberg equilibrium in cases. It is not self-evident that this is true, so it should be made clear that this is an assumption.

Formulas 776-778 should be corrected for “=0” parts.

(35) Would benefit from middle steps that say that this is Bayes rule.

Formulas (51) – (54) are not understandable as there are unexplained indexes (s and sc) and also likely something is missing.

(55) conditions on X_i = b, and right hand side has alpha_i and X_i and T (which is not defined anywhere).

I feel there is too much work left for reviewer trying to understand unclear and undefined notation and formulas. I haven’t read supplementary further.

**Have all data underlying the figures and results presented in the manuscript been provided?**

Reviewer #1: Yes

Reviewer #2: Yes

Reviewer #3: Yes

PLOS authors have the option to publish the peer review history of their article (what does this mean?). If published, this will include your full peer review and any attached files.

Reviewer #1: No

Reviewer #2: No

Reviewer #3: No

---

## [Decision Letter · Decision Letter 1]

7 Jun 2020

Dear Dr Yuan,

Thank you very much for submitting your Research Article entitled 'Leveraging correlations between variants in polygenic risk scores to detect heterogeneity in GWAS cohorts' to PLOS Genetics. Your manuscript was fully evaluated at the editorial level and by independent peer reviewers. The reviewers appreciated the attention to an important topic but identified some aspects of the manuscript that should be improved.

We therefore ask you to modify the manuscript according to the review recommendations before we can consider your manuscript for acceptance. Your revisions should address the specific points made by each reviewer.

[LINK]

Yours sincerely,

Samuli Ripatti

Associate Editor

PLOS Genetics

Scott Williams

Section Editor: Natural Variation

PLOS Genetics

Reviewer's Responses to Questions

**Comments to the Authors:**

Reviewer #2: The authors have addressed my comments.

The current version looks much better, particularly in describing the models (logistic / liability thres)

Reviewer #3: I thank the Authors for expanding the manuscript and clarified the methods considerably. The manuscript has interesting methodological ideas that are useful for the field.

Main comments:

1. Such a clearly differing behavior of correlation between predictors under logistic and probit models seems surprising. I would still suggest that the Authors check that they are simulating data from the same populations under both models.

In particular, on p.26 l.546-547 it sounds like the prevalence in logistic model has been set for the rare set of individuals who carry no risk alleles at all, and hence the population prevalence will become much larger than the prevalence in those genetically protected extreme individuals. (Unless genotypes have been mean-centered in logistic model as they seem to be in liability threshold model formula 3.) The same issue with Risch model where Authors say in their answers that “maximum possible risk is given by (0.05)*(1.16)^(2*10) = 0.97”, but with this approach the prevalence in the population would rather be

23% = sum( dbinom(0:20, size = 20, prob = 0.5) * 0.05 * 1.16^ (0:20) ) than target 5%.

Similarly you could compute the population prevalence that the logistic regression model is imposing as the expectation of probability of being case (formula 2) over the population genotype distribution.

If there are differences between population prevalence in logistic and liability threshold model simulation in Figure 1, that could affect the observed differences in SNP correlations in cases. My suggestion is that Authors check these simulations once more to make sure that both models are simulating genotype data from a population where the POPULATION PREVALENCE is 5% (or 1%) and the effect sizes are the same per allele.

2. Terminology:

Usually “probit model” means a liability threshold model with Gaussian CDF, and this model is often called simply “liability threshold model” as is done also in this manuscript. Logistic regression model is also a version of liability threshold model, although not typically called as such, but it seems confusing to me if logistic regression is now called a “probit model” as on p.7 l.151-152 and p.27 l. 564.

The end of sentence on l.152 “dot product … sigma(1-sigma)” is unclear.

3. Figure 1:

Are the colors wrong way around in the figure? Or are the results wrong way around?

In panel B, blue has larger negative bias but the legend says blue is logistic model and that in B logistic model is having "notably smaller negative bias"?

Similarly in C, blue seems Normal and cyan seems logistic?

4. l.754-755. Where is it shown that CliP score ~ N(0,1) under the null. Section 4.2. does not mention that at all.

Minor comments:

p.28 “The case probabilities conditioned on SNP values P (y=1|X j ) are calculated from the liability threshold model in equation 3.”

How do you get the effect sizes on liability scale? Refer to the equation that you use later in formula 29.

Supp. Text

L.993 “{1}^N” misses “0”

L.995 One assumption not mentioned here is that genotypes at different loci are uncorrelated in the population.

**Have all data underlying the figures and results presented in the manuscript been provided?**

Reviewer #2: Yes

Reviewer #3: Yes

PLOS authors have the option to publish the peer review history of their article (what does this mean?). If published, this will include your full peer review and any attached files.

Reviewer #2: No

Reviewer #3: No

---

## [Decision Letter · Decision Letter 2]

29 Jul 2020

Dear Dr Yuan,

We are pleased to inform you that your manuscript entitled "Leveraging correlations between variants in polygenic risk scores to detect heterogeneity in GWAS cohorts" has been editorially accepted for publication in PLOS Genetics. Congratulations!

Yours sincerely,

Samuli Ripatti

Associate Editor

PLOS Genetics

Scott Williams

Section Editor: Natural Variation

PLOS Genetics

Comments from the reviewers (if applicable):

Reviewer's Responses to Questions

**Comments to the Authors:**

Reviewer #3: Thank you for the revision. I have no further comments.

**Have all data underlying the figures and results presented in the manuscript been provided?**

Reviewer #3: Yes

PLOS authors have the option to publish the peer review history of their article (what does this mean?). If published, this will include your full peer review and any attached files.

Reviewer #3: No

**Data Deposition**

http://datadryad.org/submit?journalID=pgenetics&manu=PGENETICS-D-19-02004R2

**Press Queries**

---

## [Editor Report · Acceptance letter]

9 Sep 2020

PGENETICS-D-19-02004R2 

Leveraging correlations between variants in polygenic risk scores to detect heterogeneity in GWAS cohorts 

Dear Dr Yuan, 

We are pleased to inform you that your manuscript entitled "Leveraging correlations between variants in polygenic risk scores to detect heterogeneity in GWAS cohorts" has been formally accepted for publication in PLOS Genetics! Your manuscript is now with our production department and you will be notified of the publication date in due course.

With kind regards,

Matt Lyles

PLOS Genetics

On behalf of:
